# Multiview Self-Representation Learning across Heterogeneous Views

Jie Chen [1]   Zhu Wang [2]   Chuanbin Liu [3]   Xi Peng [4]

## Abstract

Features of the same sample generated by different pretrained models often exhibit inherently distinct feature distributions. Learning invariant representations from large-scale unlabeled visual data in a fully unsupervised transfer manner remains a significant challenge. In this paper, we propose a multiview self-representation learning (MSRL) method in which invariant representations are learned by exploiting the self-representation property of features across heterogeneous views. The features are derived from large-scale unlabeled visual data through transfer learning with various pretrained models and are referred to as heterogeneous multiview data. We introduce an information-passing mechanism that relies on self-representation learning to support feature aggregation over the outputs of the linear model. Moreover, an assignment probability distribution consistency scheme is presented to guide multiview self-representation learning by exploiting complementary information across different views. Consequently, representation invariance across different linear models is enforced through this scheme. Additionally, we provide a theoretical analysis of the assignment probability distribution consistency and the incremental views. Extensive experiments demonstrate that the proposed MSRL method consistently outperforms several state-of-the-art approaches.

## 1. Introduction

Learning invariant representations from large-scale visual data with manual annotations often yields human-level performance on downstream tasks such as image recognition (Hu et al., 2024), object detection (Hegde et al., 2025), and visual tracking (Pang et al., 2025). However, obtaining such annotations is extremely expensive and may not be necessary for deep representation learning (Gui et al., 2024). Learning invariant representations from large-scale visual data in a fully unsupervised transfer manner presents a significant challenge.

Transfer learning is a fundamental knowledge-transfer technique in which pretrained deep neural network models are used to improve the performance of the models on target data (You et al., 2021). Numerous pretrained deep neural network models can provide favorable initialization through transfer learning (Liu et al., 2022; Abuduweili et al., 2021). The invariant representations learned from large-scale visual data by fine-tuning the entire model can be effectively transferred to various downstream tasks (Darcet et al., 2024; HaoChen et al., 2022). These methods have shown significant advantages over traditional supervised learning methods. However, substantial performance improvements are observed when the entire vision model is fine-tuned on downstream tasks. In addition, such approaches still require a small number of labeled samples.

Several unsupervised transfer learning methods have been introduced to infer how samples should be labeled in downstream tasks without the need for task-specific representation learning (Gadetsky et al., 2024; Alkin et al., 2025). For example, Alkin *et al.* (Alkin et al., 2025) introduced a masked image modeling refiner method in which the later blocks of a pretrained model are refined to improve the pretraining objective. Gadetsky *et al.* (Gadetsky et al., 2024) proposed an unsupervised transfer learning method to maximize the margins of independent linear classifiers on top of frozen pretrained models. By leveraging foundation vision-language models, these fully unsupervised transfer methods often achieve human-level performance on downstream tasks and outperform fine-tuning-based approaches (Chen et al., 2025a; Mirza et al., 2023). However, a significant limitation of these methods lies in the effective capture of complementary information across multiple pretrained models when features are derived from visual data. Complementary information provides additional semantic knowledge preserved across different models. Therefore, learning invariant representations from different pretrained

---

[1]College of Computer Science, Sichuan University, Chengdu, China [2]Law School, Sichuan University, Chengdu, China [3]School of Economics and Management, China University of Petroleum (Beijing), Beijing, China [4]School of Artificial Intelligence, Sichuan University, Chengdu, China. Correspondence to: Xi Peng <pengx.gm@gmail.com>.

*Proceedings of the 43$^{rd}$ International Conference on Machine Learning*, Seoul, South Korea. PMLR 306, 2026. Copyright 2026 by the author(s).

models requires further investigation into how to effectively exploit complementary information across multiple views.

In recent years, contrastive learning has emerged as the dominant paradigm for unsupervised visual representation learning (Du et al., 2024; Tian et al., 2024; HaoChen et al., 2022). Contrastive learning aims to learn representations by maximizing a lower bound on the mutual information between the augmented views of visual data, where these augmented views provide complementary information. The objective losses used in contrastive learning techniques, such as InfoNCE (Oord et al., 2019) and Deep InfoMax (Hjelm et al., 2021), explicitly increase the separation between dissimilar pairs of the augmented views while reducing the separation between similar pairs. Many contrastive learning-based methods leverage the generalizability of contrastive learning to obtain compact representations without the need for manually assigned labels (Huang et al., 2024; Wang & Isola, 2020; Tian et al., 2020). These methods rapidly reduce the performance gap relative to that of supervised learning in large-scale visual data. A pair of augmented views of the same sample is encoded by the same contrastive learning-based model, resulting in representations that follow a shared underlying distribution (Chen et al., 2020). However, features of the same sample generated by different pretrained models may originate from distinct underlying distributions. Therefore, there is an urgent need to learn invariant representations from large-scale unlabeled visual data across heterogeneous views with pretrained models.

In this paper, we propose a multiview self-representation learning (MSRL) method that learns invariant representations from large-scale unlabeled visual data in a fully unsupervised transfer manner. The features are derived from the visual data through transfer learning with different pretrained models. These features are referred to as heterogeneous multiview data. An individual linear model is stacked on top of the corresponding frozen pretrained backbone. We introduce an information-passing mechanism in which self-representation learning is used to aggregate features output by the linear model. Specifically, self-representation learning exploits the self-representation property of features to define a feature aggregation operator that adaptively selects neighborhoods of varying sizes while maintaining the linear weights shared among the features. Each representation corresponding to a feature can be represented by the linear combination of its spatial neighbors. To learn invariant representations, we present an assignment probability distribution consistency scheme to guide multiview self-representation learning. The assignment probability distributions produced by the linear models contain complementary information across different views, which can provide additional semantic knowledge for clustering assignments. By exploiting complementary information across different views, representation invariance across different

views is enforced through the assignment probability distribution consistency scheme. Additionally, we provide a theoretical analysis of the assignment probability distribution consistency and the incremental views. Extensive experiments show that the proposed MSRL method consistently outperforms state-of-the-art approaches across multiple benchmark visual datasets.

The contributions of this work are summarized as follows.

- An MSRL model is introduced to learn invariant representations from multiple views of large-scale visual data in a fully unsupervised transfer manner.

- An information-passing mechanism is developed to adaptively aggregate feature information. The most linearly related features in the same category are adaptively selected to yield discriminative representations.

- An assignment probability distribution consistency scheme is introduced for multiview self-representation learning, which exploits additional semantic knowledge to effectively enforce representation invariance across different views.

## 2. Related Work

### 2.1. Unsupervised Transfer Learning

In transfer learning, fine-tuning a pretrained model for new downstream tasks has attracted considerable attention because of its demonstrated efficiency and generalizability (Abuduweili et al., 2021). However, recent studies have shown that fine-tuning an entire pretrained model on downstream tasks usually leads to faster convergence but results in only marginal improvements over unsupervised transfer learning (Gadetsky et al., 2024; HaoChen et al., 2022). Unsupervised transfer learning has become a fundamental paradigm in unsupervised visual representation learning, in which pretrained models are employed to enhance performance on target-specific downstream tasks without the need for labeled samples (Alkin et al., 2025; Chen et al., 2025a; Mirza et al., 2023). Numerous fully unsupervised transfer methods can be divided into two main categories: training a linear classifier on representations generated by frozen pretrained models and parameter-efficient transfer learning techniques that adapt pretrained models to downstream tasks while a limited set of parameters is trained (Alkin et al., 2025; Xin et al., 2024). Performance improvements in these methods arise from facilitating the adaptation of pretrained models to downstream tasks.

### 2.2. Multiview Representation Learning

The self-representation property of data samples can be employed to evaluate the membership among data samples if

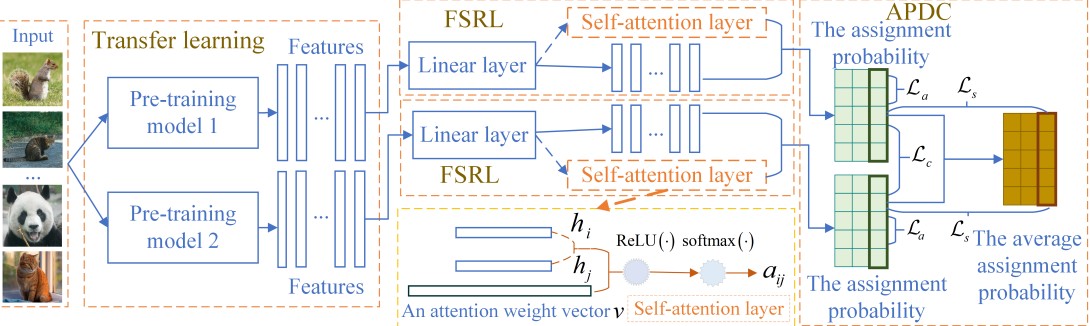

*Figure 1.* Framework of the MSRL model with two pretrained models. The framework consists of three main modules: a transfer learning module, a feature self-representation learning (FSRL) module and an assignment probability distribution consistency (APDC) module.

a linear relationship exists among data samples from the same category (Chen et al., 2025b; 2022). Specifically, a sample can be represented as a linear combination of its neighboring samples from the same category. For example, Chen *et al.* proposed an approximated local linear representation to learn sparse representations (Chen et al., 2023b). Multiview data typically provide both consistent and complementary information across multiple views. A variety of multiview representation learning methods leverage the self-representation property of data samples to learn a shared representation across multiple views (Fu et al., 2022; Chen et al., 2022). However, these traditional shallow self-representation models are inadequate for learning discriminative features from large-scale datasets. In contrast, deep self-representation learning models, such as deep autoencoders (Liu et al., 2025; Chen et al., 2023a) and graph neural networks (Fu et al., 2025; Deng et al., 2025; Veličković et al., 2018), exhibit a strong ability to capture intrinsic features within large-scale data. These studies indicate that integrating self-representation learning with deep multiview learning paradigms is crucial for effectively capturing the intrinsic structure of large-scale visual data.

# 3. Multiview Self-Representation Learning via Unsupervised Transfer Learning

## 3.1. Problem Formulation

Given a set of $L$ pretrained models $\{\phi_l\}_{l=1}^L$ and an unlabeled training dataset $\mathcal{D}_t = \{\mathbf{x}_1, \mathbf{x}_2, ..., \mathbf{x}_n\}$ consisting of $n$ samples, each pretrained model $\phi_l(\cdot)$ transforms a raw image $\mathbf{x}_i \in \mathbb{R}^m$ into a fixed-dimensional vector $\widetilde{\mathbf{x}}_i \in \mathbb{R}^d$ through transfer learning. For a given pretrained model $\phi_l$, a feature self-representation learning (FSRL) model is stacked on top of the fixed-dimensional vectors produced by the frozen pretrained model. Specifically, an FSRL model is denoted by $\delta^{(l)}(\mathbf{W}_l; \phi_l(\mathbf{x}_i))$, where $\mathbf{W}_l$ represents the trainable parameters of the FSRL model. Different pretrained models capture diverse aspects of visual data. Features of the same

raw image produced by different pretrained models often exhibit fundamentally distinct feature distributions. The critical challenge lies in learning invariant representations across heterogeneous views from large-scale visual data, while leveraging these pretrained models in a fully unsupervised manner. For simplicity, we consider two pretrained models, $\phi_u$ and $\phi_v$. The parameters $\mathbf{W}_u$ and $\mathbf{W}_v$ of the two FSRL models $\delta^{(u)}(\cdot; \cdot)$ and $\delta^{(v)}(\cdot; \cdot)$ are jointly learned by the following optimization problem:

$$\min_{\mathbf{W}_u, \mathbf{W}_v} \frac{1}{n} \sum_{r=1}^n \mathcal{L}\left(\delta^{(u)}(\mathbf{W}_u; \phi_u(\mathbf{x}_r)), \delta^{(v)}(\mathbf{W}_v; \phi_v(\mathbf{x}_r))\right) \tag{1}$$

where $\mathcal{L}$ represents a general loss function.

## 3.2. Network Architecture

The network architecture of the MSRL model is illustrated in Fig. 1. It consists of three main modules: a transfer learning module, a feature self-representation learning module and an assignment probability distribution consistency module. The transfer learning module incorporates multiple pretrained models to extract features that capture high-level semantic information from visual data, which results in heterogeneous multiview features. Given these features, the MSRL framework jointly leverages the self-representation property of these features to learn invariant representations in an unsupervised transfer learning manner.

## 3.3. Information-Passing Mechanism for Feature Self-Representation Learning

Given sample $\mathbf{x}_i$, a low-dimensional feature $\mathbf{h} \in \mathbb{R}^C$ is obtained by applying a linear model $\sigma(\cdot; \cdot)$ to the output of a pretrained model $\phi(\mathbf{x}_i)$, as follows:

$$\mathbf{h} = \sigma\left(\mathbf{W}^\top; \phi(\mathbf{x}_i)\right) \tag{2}$$

where $\mathbf{W} \in \mathbb{R}^{d \times C}$ represents the trainable parameters of the linear model, $C$ denotes the number of clusters, and

$\sigma\left(\cdot;\cdot\right)$ denotes a linear transformation function. Let $\mathbf{H} = [\mathbf{h}_1, \mathbf{h}_2, ..., \mathbf{h}_n] \in \mathbb{R}^{C \times n}$ represent a set of low-dimensional features generated using Eq. (2). The basic assumption on local smoothness is that the features $\mathbf{H}$ produced by the linear model $\sigma\left(\cdot;\cdot\right)$ can be partitioned into $C$ clusters by measuring local similarity with each other.

**Assumption 3.1** (Local Smoothness). If two features $\mathbf{h}_i$ and $\mathbf{h}_j$ are close to each other, i.e., $\mathbf{h}_j \in \mathbb{N}_k\left(\mathbf{h}_i\right)$, in a smooth low-dimensional manifold, their soft cluster assignments are likely to be similar, where $\mathbb{N}_k\left(\mathbf{h}_i\right)$ denotes the set of the $k$-nearest neighbors of $\mathbf{h}_i$.

The features $\mathbf{h}_i$ and $\mathbf{h}_j$ $(1 \le i, j \le n)$ within the same cluster are linearly dependent. By leveraging the self-representation property of these features, we adopt $\mathbf{H}$ as the new basis matrix. Thus, each feature $\mathbf{h}_i$ $(1 \le i \le n)$ can be represented by a linear combination of $\mathbf{H}$ as follows:

$$\mathbf{h}_i = \mathbf{H}\mathbf{a}_i + \mathbf{e}_i \qquad (3)$$

where $\mathbf{a}_i \in \mathbb{R}^n$ is a coefficient vector, and $\mathbf{e}_i \in \mathbb{R}^C$ is an error term. FSRL as defined in Eq. (3) is a fundamental self-representation formulation of MSRL. From the perspective of graph theory (Belkin & Niyogi, 2003), if two features $\mathbf{h}_i$ and $\mathbf{h}_j$ are proximate in the intrinsic geometry of the data distribution, their corresponding representations $\mathbf{a}_i$ and $\mathbf{a}_j$ are expected to be close to each other under an appropriately defined basis. This relationship indicates that $\mathbf{a}_i$ and $\mathbf{a}_j$ are proximate if $\mathbf{h}_i$ and $\mathbf{h}_j$ belong to the same cluster.

Each coefficient $a_{ij}$ in $\mathbf{a}_i$ is closely associated with the neighbors of the feature $\mathbf{h}_i$. To obtain an appropriate coefficient vector $\mathbf{a}_i$, we introduce an information-passing mechanism in MSRL. Specifically, information is aggregated such that a representation corresponding to each feature is represented by its spatial neighbors, and their information is leveraged for self-representation. Given a feature $\mathbf{h}$, the challenge lies in defining an operator that adaptively selects neighborhoods of varying sizes while maintaining the linear weights shared among the features in $\mathbf{H}$ within the information-passing mechanism. An attention mechanism is adopted to capture the relationships among the features in $\mathbf{H}$, emphasizing the most linearly related features within the same category to produce discriminative representations (Veličković et al., 2018; Vaswani et al., 2017). The coefficients $a_{ij}$ between the features $\mathbf{h}_i$ and $\mathbf{h}_j$ computed by the attention mechanism are formulated as follows:

$$a_{ij} = \frac{\exp\left(\mathrm{ReLU}\left(\mathbf{v}^\top [\mathbf{h}_i \parallel \mathbf{h}_j]\right)\right)}{\sum\limits_{r=1}^{B} \exp\left(\mathrm{ReLU}\left(\mathbf{v}^\top [\mathbf{h}_i \parallel \mathbf{h}_r]\right)\right)} \qquad (4)$$

where $\mathbf{v} \in \mathbb{R}^{2C}$ is a weight vector learned by a single linear layer, $B$ is the batch size, and $[\cdot \parallel \cdot]$ denotes the concatenation operation of two features. The attention mechanism in

Eq. (4) provides a concrete implementation for learning the coefficient vector of the self-representation formulation.

The information-passing mechanism allows the representation $\mathbf{z}_i \in \mathbb{R}^C$, corresponding to the sample $\mathbf{x}_i$, to be formulated as a linear combination of features from the same category, i.e., by aggregating information from spatially proximate neighbors. The aggregation operation of $\mathbf{h}_i$, i.e., $f\left(\mathbf{h}_i; \mathbf{a}_i\right)$, is defined as follows:

$$\mathbf{z}_i = f\left(\mathbf{h}_i; \mathbf{a}_i\right) = \sum_{j=1}^{B} a_{ij}\mathbf{h}_j + \mathbf{h}_i. \qquad (5)$$

Through the information-passing mechanism, MSRL aggregates information from neighboring features within the same category to central features, which allows it to effectively capture the geometric structure of these features. The information-passing mechanism provides an interpretive perspective for the mathematical formulation of FSRL. Given a pretrained model, the assignment probability distribution of the representation $\mathbf{z}_i$, i.e., $\mathbf{s}_i \in \Delta^{C-1}$, corresponding to sample $\mathbf{x}_i$ $(1 \le i \le n)$, is calculated as follows:

$$\mathbf{s}_i = \delta\left(\mathbf{z}_i\right) \qquad (6)$$

where $\delta\left(\cdot\right)$ is a softmax activation function and $\Delta^{C-1}$ denotes the $(C-1)$-dimensional probability simplex, i.e., $\Delta^{C-1} = \left\{\mathbf{w} \in \mathbb{R}^C : w_c \ge 0, \sum\limits_{c=1}^{C} w_c = 1\right\}$.

### 3.4. Assignment Probability Distribution Consistency among Multiple Views

Different pretrained models typically yield features with heterogeneous feature distributions for the same sample. These features are referred to as multiview data. Since they often follow heterogeneous feature distributions, applying contrastive learning directly to them is not suitable for the MSRL model. To overcome this limitation, we present an assignment probability distribution consistency scheme for multiview self-representation learning. The assignment probability distributions produced by the linear models provide additional semantic knowledge for cluster assignments. In particular, these assignment probability distributions contain complementary information across different views.

The linear models corresponding to the pretrained models are expected to yield consistent cluster assignments. Given $L$ pretrained models, a latent semantic assignment probability distribution for a sample $\mathbf{x}_i$ across the linear models is calculated as follows:

$$\rho_\theta\left(\mathbf{x}_i\right) = \frac{1}{L}\sum_{l=1}^{L} \mathbf{s}_i^{(l)} \qquad (7)$$

where $\mathbf{s}_i^{(l)} = \delta_l\left(f_l\left(\sigma_l\left((\mathbf{W}_l)^\top; \phi_l\left(\mathbf{x}_i\right)\right); \mathbf{a}_i^{(l)}\right)\right)$. The

cluster assignment of sample $\mathbf{x}_i$ can be predicted by

$$y_i = \arg\max_j \left[\rho_\theta\left(\mathbf{x}_i\right)\right]_j, \quad j \in \{1, 2, ..., C\} \quad (8)$$

where $\left[\rho_\theta\left(\mathbf{x}_i\right)\right]_j$ represents the assignment probability distribution of a sample $\mathbf{x}_i$ to the $j$th cluster. The cluster assignment provides specific semantic knowledge for generating pseudolabels. Under the guidance of these assignments, the semantic pseudolabel loss for unlabeled samples is formulated as follows:

$$\mathcal{L}_s = \frac{1}{B} \sum_{i=1}^{B} \frac{1}{L} \sum_{l=1}^{L} \mathcal{H}\left(y_i, \mathbf{s}_i^{(l)}\right) \quad (9)$$

where $\mathcal{H}\left(\cdot, \cdot\right)$ is the cross-entropy loss, i.e., $\mathcal{H}\left(y_i, \mathbf{s}_i^{(l)}\right) = -\log \mathbf{s}_{i,y_i}^{(l)}$. The assignment probability distributions produced by all the linear models $\left\{\mathbf{s}_i^{(l)}\right\}_{l=1}^{L}$ for sample $\mathbf{x}_i$ are directed toward the same probability direction, which indicates that these assignment probability distributions are jointly attracted toward the consensus $\rho_\theta\left(\mathbf{x}_i\right)$. Therefore, the consensus distribution is a stable attractor for all linear models.

To avoid all samples being assigned to a single cluster, we introduce the following regularization term:

$$\mathcal{L}_a = \sum_{l=1}^{L} \sum_{j=1}^{C} q_j^{(l)} \log q_j^{(l)} \quad (10)$$

where $q_j^{(l)}$ is defined as $q_j^{(l)} = \frac{1}{B} \sum_{i=1}^{B} s_{ij}^{(l)}$. This term is considered a cluster diversity loss in the MSRL model (Xu et al., 2022), which promotes balanced cluster assignments by maximizing the marginal entropy of the probability assignment for each FSRL component.

The consistency loss is an important component widely used in various self-supervised learning algorithms. The features from different views tend to share consistent assignment probability distributions. By minimizing the discrepancy among the assignment probability distributions produced by multiple linear models, the generalizability of the linear models is enhanced. The cross-view consistency loss over assignment probability distributions is defined as follows:

$$\mathcal{L}_c = \frac{1}{B} \sum_{i=1}^{B} \sum_{p=1}^{L} \sum_{q=1, p \neq q}^{L} \mathcal{H}\left(\mathbf{s}_i^{(p)}, \mathbf{s}_i^{(q)}\right) \quad (11)$$

where $\mathbf{s}_i^{(p)}$ and $\mathbf{s}_i^{(q)}$ are the assignment probability distributions produced by the $p$th and $q$th linear models, respectively. The cross-entropy between $\mathbf{s}_i^{(p)}$ and $\mathbf{s}_i^{(q)}$ is given by

$$\mathcal{H}\left(\mathbf{s}_i^{(p)}, \mathbf{s}_i^{(q)}\right) = -\sum_{j=1}^{C} s_{ij}^{(p)} \log s_{ij}^{(q)}. \quad (12)$$

Eq. (12) drives the assignments of different views for each sample to converge in the probability simplex, which leads to refined alignment across multiple views.

The MSRL model employs three complementary loss components, i.e., the semantic pseudolabel loss, the cluster diversity loss and the cross-view consistency loss, that jointly enforce assignment probability distribution consistency among the linear models. The overall loss objective of the MSRL model is given as follows:

$$\mathcal{L} = \mathcal{L}_s + \alpha \mathcal{L}_a + \beta \mathcal{L}_c \quad (13)$$

where $\alpha$ and $\beta$ are tradeoff hyperparameters. The assignment probability distributions produced by different linear models associated with pretrained backbones converge to a unified distribution. The entire optimization procedure of the proposed MSRL method is included in Appendix A.

### 3.5. Theoretical Justification

In this section, we provide theoretical discussions to clarify multiview self-representation learning and its connections with the assignment probability distribution consistency and incremental views. Additionally, proofs for all the theorems are available in the Appendix B.

#### 3.5.1. ASSIGNMENT PROBABILITY DISTRIBUTION CONSISTENCY

Let $\left\{\mathbf{s}_i^{(l)}\right\}_{l=1}^{L} \in \Delta_\delta^{C-1}$ denote the assignment probability distributions produced by the $L$ linear models for sample $\mathbf{x}_i$, where $\Delta_\delta^{C-1}$ is the probability simplex, i.e., $\Delta_\delta^{C-1} = \left\{\mathbf{w} \in \Delta^{C-1} : w_c \geq \delta > 0\right\}$. As $\mathbf{s}_i^{(l)}$ is generated by a softmax operator in Eq. (6), it lies in the interior of the probability simplex. In particular, the entropy function $\mathcal{H}\left(\cdot\right)$ is Lipschitz-continuous in this domain. For fixed $\mathbf{s}_i^{(p)}$, the cross-entropy $\mathcal{H}\left(\mathbf{s}_i^{(p)}, \mathbf{s}_i^{(q)}\right)$ in Eq. (11) is strictly convex in $\mathbf{s}_i^{(q)}$ and uniquely minimized when $\mathbf{s}_i^{(p)} = \mathbf{s}_i^{(q)}$.

**Theorem 3.2** (Bounded Multiview Consistency). *Considering the sample $\mathbf{x}_i$, assume that there exists a sufficiently small constant $\varepsilon > 0$ such that $\mathbb{E}\left[\left\|\mathbf{s}_i^{(p)} - \mathbf{s}_i^{(q)}\right\|_1\right] \leq \varepsilon, \quad \forall p \neq q$, where $\mathbf{s}_i^{(p)} \in \Delta_\delta^{C-1}$ and $\mathbf{s}_i^{(q)} \in \Delta_\delta^{C-1}$ denote the assignment probability distributions produced by the $p$th and $q$th linear models, respectively. Then, there exists a latent semantic assignment probability distribution $\mathbf{p}_i^{(L)} \in \Delta_\delta^{C-1}$ for sample $\mathbf{x}_i$ over $L$ linear models, defined as $\mathbf{p}_i^{(L)} = \rho_\theta\left(\mathbf{x}_i\right)$, which satisfies the following properties:*

(1) $\mathbb{E}\left[\mathcal{H}\left(\mathbf{p}_i^{(L)}\right)\right] \leq \frac{1}{L} \sum_{l=1}^{L} \mathbb{E}\left[\mathcal{H}\left(\mathbf{s}_i^{(l)}\right)\right] + Q_\delta \varepsilon;$

(2) $\mathbb{E}\left[\left\|\mathbf{s}_i^{(l)} - \mathbf{p}_i^{(L)}\right\|_1\right] \leq \varepsilon, \quad \forall l = 1, \ldots, L$

*where $L$ denotes the number of pretrained models, and $Q_\delta > 0$ is the Lipschitz constant of the entropy function on the probability simplex $\Delta_\delta^{C-1}$.*

The semantic pseudolabel loss in Eq. (9) is minimized when the assignment probability distribution is concentrated on the same category $y_i$ for all pretrained models, where $y_i$ is computed from $\rho_\theta(\mathbf{x}_i)$. Theorem 3.2 establishes two fundamental properties of assignment probability distribution consistency. Specifically, information fusion of assignment probability distributions across multiple views is effectively achieved in the probability simplex by averaging assignment probability distributions. Moreover, under the assumption that the disagreement among view-specific assignment probability distributions is bounded, the entropy of the fused distribution remains bounded within $Q_\delta \varepsilon$ above the average entropy of the individual views, providing a theoretical guarantee of the fused assignment probability distribution. Therefore, the assignment probability distributions $\mathbf{s}_i^{(l)}$ of all linear models converge to the latent semantic assignment distribution $\mathbf{p}_i^{(L)}$ for sample $\mathbf{x}_i$.

### 3.5.2. INCREMENTAL VIEW ANALYSIS

We present a view-incremental theoretical analysis to characterize how clustering quality evolves as heterogeneous views corresponding to different pretrained models are progressively introduced into MSRL. Specifically, when a new pretrained model is incorporated into an existing collection of $L$ pretrained models, the latent semantic assignment probability distribution $\mathbf{p}_i^{(L+1)}$ for a sample $\mathbf{x}_i$ over the $(L+1)$ linear models is updated in an incremental manner. Such incrementally incorporated heterogeneous views define an incremental consensus dynamical system. According to Eq. (7), the latent semantic assignment distribution $\mathbf{p}_i^{(L+1)}$ for sample $\mathbf{x}_i$ is updated incrementally as follows:

$$\mathbf{p}_i^{(L+1)} = \frac{L}{L+1}\mathbf{p}_i^{(L)} + \frac{1}{L+1}\mathbf{s}_i^{(L+1)} \qquad (14)$$

where $\mathbf{p}_i^{(L)}$ is defined as $\mathbf{p}_i^{(L)} = \rho_\theta(\mathbf{x}_i)$ with $L$ pretrained models and $\mathbf{s}_i^{(L+1)}$ denotes the assignment probability distribution produced by the $(L+1)$th linear model. This defines an incremental updating rule for the latent semantic assignment probability distribution.

**Assumption 3.3** (Bounded Multiview Discrepancy). Given a sample $\mathbf{x}_i$, assume that there exists a sufficiently small constant $\varepsilon > 0$ such that, for each view $l$,

$$\mathbb{E}\left[\left\|\mathbf{s}_i^{(l)} - \mathbf{p}_i\right\|_1\right] \leq \varepsilon, \quad \forall l \geq 1$$

where $\mathbf{p}_i \in \Delta_\delta^{C-1}$ denotes a consensus assignment probability distribution, $\mathbf{s}_i^{(l)} \in \Delta_\delta^{C-1}$ represents the assignment probability distribution produced by the $l$th linear model, and $L$ denotes the number of pretrained models.

**Theorem 3.4** (Attractivity of Consensus Distribution). *Under Assumption 3.3, there exists a constant $w > 0$ such that*

$$\mathbb{E}\left[\left\|\mathbf{p}_i^{(L)} - \mathbf{p}_i\right\|_1\right] \leq w\left(\frac{1}{L} + \varepsilon\right)$$

*where $\mathbf{p}_i^{(L)}$ is a latent semantic assignment probability distribution for the sample $\mathbf{x}_i$ over $L$ linear models.*

Theorem 3.4 indicates that the latent semantic assignment probability distribution $\mathbf{p}_i^{(L)}$ is attracted to the consensus assignment probability distribution $\mathbf{p}_i$ as $L$ increases. In particular, the effect of stochastic perturbations remains bounded by $\mathcal{O}(\varepsilon)$. Consequently, $\mathbf{p}_i$ is regarded as a stable attractor in expectation for an incremental consensus dynamic system. This attractor property explains why the latent semantic assignment probability distribution $\mathbf{p}_i^{(L)}$ stabilizes as $L$ increases under Assumption 3.3. In addition, Theorem 3.4 characterizes how uncertainty evolves along the attracting trajectory.

**Theorem 3.5** (Bounded Entropy Variation under Incremental Views). *Under Assumption 3.3, the expected entropy of $\mathbf{p}_i^{(L+1)}$ for the sample $\mathbf{x}_i$ satisfies*

$$\mathbb{E}\left[\mathcal{H}(\mathbf{p}_i^{(L+1)})\right] - \mathbb{E}\left[\mathcal{H}(\mathbf{p}_i^{(L)})\right] \leq \frac{2Q_\delta \varepsilon}{L+1},$$

*where $\mathbf{p}_i^{(L)}$ is a consensus assignment probability distribution for the sample $\mathbf{x}_i$ over the $L$ linear models, and $Q_\delta > 0$ is the Lipschitz constant of the entropy function on the probability simplex $\Delta_\delta^{C-1}$.*

Theorem 3.5 characterizes the entropy evolution of the consensus assignment probability distribution under incremental heterogeneous views. Specifically, when the assignment probability distributions produced by individual linear models are mutually consistent up to a bounded deviation, the change in entropy induced by incorporating an additional view is tightly controlled. This result also indicates that not every additional view is guaranteed to be beneficial. In addition, Theorem 3.5 establishes that the expected entropy variation decays at a rate of $\mathcal{O}(\varepsilon/(L+1))$. This implies that as more heterogeneous views are aggregated, the consensus assignment probability distribution becomes increasingly stable in terms of uncertainty, and any potential increase in entropy vanishes asymptotically. From the perspective of the incremental consensus dynamical system, this result indicates that the incremental update on the consensus assignment probability distribution forms a stabilizing process in the probability simplex, which prevents entropy inflation caused by heterogeneous views.

*Table 1.* Clustering performance of the different methods on eight vision datasets.

| Methods | Pets ACC | NMI | ARI | GTSRB ACC | NMI | ARI | DTD ACC | NMI | ARI | Aircraft ACC | NMI | ARI | Flowers ACC | NMI | ARI | CIFAR-10 ACC | NMI | ARI | CIFAR-100 ACC | NMI | ARI | ImageNet-1K ACC | NMI | ARI |
|---|---|---|---|---|---|---|---|---|---|---|---|---|---|---|---|---|---|---|---|---|---|---|---|---|
| CMVC | 90.30 | 92.29 | 84.20 | 31.12 | 57.45 | 25.17 | 54.36 | 66.27 | 39.06 | 32.31 | 65.23 | 24.63 | 77.75 | 94.28 | 81.91 | 99.14 | 97.68 | 98.11 | 57.43 | 78.66 | 49.08 | - | - | - |
| SparseMVC | 87.95 | 94.81 | 86.75 | 35.65 | 59.15 | 32.98 | 52.55 | 66.91 | 38.98 | 24.87 | 62.74 | 23.79 | 89.15 | 96.88 | 89.42 | 97.86 | 94.23 | 95.31 | 42.14 | 73.74 | 37.00 | 14.44 | 63.73 | 7.46 |
| HSACC | 84.86 | 92.24 | 82.37 | 33.39 | 56.47 | 27.86 | 51.81 | 63.11 | 37.93 | 25.05 | 53.91 | 16.60 | 96.30 | 98.71 | 96.25 | 92.55 | 93.99 | 88.77 | 64.36 | 81.44 | 56.61 | - | - | - |
| BONE | 92.75 | 94.21 | 88.87 | 40.91 | 61.24 | 36.31 | 58.04 | 68.53 | 43.32 | 32.22 | 64.04 | 25.72 | 94.71 | 98.82 | 96.87 | 99.41 | 98.32 | 98.71 | 87.39 | 90.55 | 80.02 | 68.01 | 83.82 | 55.63 |
| CPP | 91.19 | 85.09 | 82.29 | 43.23 | 63.90 | 36.15 | 57.94 | 55.29 | 41.68 | 34.85 | 57.04 | 23.78 | 93.29 | 88.59 | 86.69 | 97.94 | 94.90 | 94.10 | 76.26 | 83.65 | 72.05 | 67.48 | 85.16 | 54.57 |
| CLUDI | 89.55 | 84.43 | 80.50 | 38.49 | 60.74 | 33.50 | 56.33 | 51.76 | 39.89 | 36.88 | 63.47 | 26.20 | 86.78 | 83.93 | 82.63 | 91.06 | 93.14 | 90.65 | 85.36 | 87.22 | 79.16 | 60.81 | 81.69 | 52.10 |
| MIM-Refiner | 83.89 | 87.89 | 76.49 | 33.63 | 52.89 | 27.50 | 50.69 | 58.22 | 33.12 | 24.61 | 45.48 | 15.37 | 97.50 | 77.77 | 88.33 | 91.09 | 94.51 | 94.93 | 75.38 | 81.00 | 62.73 | 63.24 | 84.72 | 40.18 |
| TURTLE | 92.59 | 93.80 | 87.62 | 45.01 | 65.02 | 36.89 | 57.55 | 68.42 | 42.69 | 36.78 | 60.80 | 25.76 | 99.56 | 99.68 | 99.48 | 99.50 | 98.56 | 98.90 | 89.10 | 91.35 | 82.46 | 70.10 | 86.58 | 58.42 |
| MSRL | 95.83 | 95.83 | 92.14 | 49.78 | 65.89 | 46.65 | 62.34 | 72.89 | 49.03 | 37.65 | 66.96 | 27.72 | 99.71 | 99.76 | 99.58 | 99.57 | 98.74 | 99.05 | 91.00 | 92.34 | 84.72 | 73.24 | 88.26 | 62.80 |

*Table 2.* Clustering performance of MSRL using different combinations of pretrained models across eight vision datasets.

| Model | L/14 | B/16 | B/32 | Pets ACC | NMI | ARI | GTSRB ACC | NMI | ARI | DTD ACC | NMI | ARI | Aircraft ACC | NMI | ARI | Flowers ACC | NMI | ARI | CIFAR-10 ACC | NMI | ARI | CIFAR-100 ACC | NMI | ARI | ImageNet-1K ACC | NMI | ARI |
|---|---|---|---|---|---|---|---|---|---|---|---|---|---|---|---|---|---|---|---|---|---|---|---|---|---|---|---|
| DINOv2 | ✓ | | | 95.83 | 95.83 | 92.14 | 49.78 | 65.89 | 46.65 | 62.34 | 72.89 | 49.03 | 37.65 | 66.96 | 27.72 | 99.71 | 99.76 | 99.58 | 99.57 | 98.74 | 99.05 | 91.00 | 92.34 | 84.72 | 73.24 | 88.26 | 62.80 |
| | | ✓ | | 93.35 | 94.41 | 89.48 | 41.65 | 58.80 | 36.01 | 62.66 | 72.44 | 49.08 | 33.54 | 64.42 | 24.53 | 99.74 | 99.79 | 99.64 | 99.41 | 98.27 | 98.69 | 88.43 | 90.78 | 80.98 | 71.46 | 87.30 | 60.25 |
| | | | ✓ | 93.30 | 94.19 | 88.36 | 41.16 | 58.92 | 33.93 | 60.74 | 70.79 | 46.60 | 35.61 | 65.92 | 26.75 | 99.76 | 99.80 | 99.66 | 99.37 | 98.18 | 98.61 | 86.79 | 89.95 | 79.05 | 71.05 | 87.15 | 59.78 |
| | ✓ | ✓ | | 96.46 | 96.47 | 93.60 | 47.02 | 66.52 | 42.01 | 64.95 | 75.21 | 52.57 | 40.35 | 70.03 | 31.57 | 99.79 | 99.82 | 99.72 | 99.47 | 98.45 | 98.83 | 89.60 | 91.44 | 82.88 | 72.97 | 88.11 | 62.30 |
| | ✓ | | ✓ | 94.96 | 95.53 | 91.12 | 45.96 | 64.57 | 41.31 | 64.36 | 74.64 | 52.10 | 40.47 | 69.21 | 31.72 | 99.76 | 99.80 | 99.68 | 99.46 | 98.40 | 98.81 | 89.23 | 91.35 | 82.47 | 72.40 | 87.69 | 61.31 |
| | ✓ | ✓ | ✓ | 95.50 | 95.45 | 91.54 | 47.92 | 69.02 | 41.41 | 66.22 | 75.68 | 53.97 | 39.24 | 68.26 | 30.58 | 99.76 | 99.81 | 99.69 | 99.40 | 98.25 | 98.67 | 88.71 | 90.86 | 81.24 | 70.46 | 87.12 | 59.73 |

*Table 3.* Ablation study of the main components in Eq. (13) with eight vision datasets.

| Methods | $\mathcal{L}_s$ | $\mathcal{L}_a$ | $\mathcal{L}_c$ | Pets ACC | NMI | ARI | GTSRB ACC | NMI | ARI | DTD ACC | NMI | ARI | Aircraft ACC | NMI | ARI | Flowers ACC | NMI | ARI | CIFAR-10 ACC | NMI | ARI | CIFAR-100 ACC | NMI | ARI | ImageNet-1K ACC | NMI | ARI |
|---|---|---|---|---|---|---|---|---|---|---|---|---|---|---|---|---|---|---|---|---|---|---|---|---|---|---|---|
| $MSRL_{avg}$ | ✓ | ✓ | | 90.05 | 92.73 | 85.27 | 40.00 | 56.25 | 29.79 | 54.47 | 68.31 | 42.30 | 29.25 | 57.66 | 20.89 | 99.20 | 99.64 | 99.42 | 99.37 | 98.17 | 98.61 | 84.82 | 88.82 | 77.22 | 72.31 | 88.27 | 62.47 |
| MSRL | ✓ | ✓ | ✓ | 95.83 | 95.83 | 92.14 | 49.78 | 65.89 | 46.65 | 62.34 | 72.89 | 49.03 | 37.65 | 66.96 | 27.72 | 99.71 | 99.76 | 99.58 | 99.57 | 98.74 | 99.05 | 91.00 | 92.34 | 84.72 | 73.24 | 88.26 | 62.80 |

*Table 4.* Comparison between $MSRL_v$ and MSRL.

| Methods | Pets ACC | NMI | ARI | GTSRB ACC | NMI | ARI | DTD ACC | NMI | ARI | Aircraft ACC | NMI | ARI | Flowers ACC | NMI | ARI | CIFAR-10 ACC | NMI | ARI | CIFAR-100 ACC | NMI | ARI | ImageNet-1K ACC | NMI | ARI |
|---|---|---|---|---|---|---|---|---|---|---|---|---|---|---|---|---|---|---|---|---|---|---|---|---|
| $MSRL_v$ | 93.21 | 94.31 | 88.94 | 42.63 | 63.85 | 34.88 | 60.32 | 70.48 | 45.89 | 36.42 | 65.20 | 27.79 | 99.54 | 99.67 | 99.47 | 99.50 | 98.51 | 98.89 | 90.21 | 91.86 | 83.57 | 71.93 | 87.87 | 61.43 |
| MSRL | 95.83 | 95.83 | 92.14 | 49.78 | 65.89 | 46.65 | 62.34 | 72.89 | 49.03 | 37.65 | 66.96 | 27.72 | 99.71 | 99.76 | 99.58 | 99.57 | 98.74 | 99.05 | 91.00 | 92.34 | 84.72 | 73.24 | 88.26 | 62.80 |

# 4. Experiments

## 4.1. Experimental Settings

Following the experimental settings in the previous work (Gadetsky et al., 2024), two representative pretrained models, i.e., DINOv2 (Oquab et al., 2024) and CLIP ViT-L/14 (Radford et al., 2021), are employed as backbones to derive two heterogeneous feature views. We evaluate the clustering performance of MSRL on eight publicly available vision datasets, namely, the Pets, GTSRB, DTD, Aircraft, Flowers, CIFAR-10, CIFAR-100 and ImageNet-1K datasets. The statistics of these datasets and the parameter settings are included in the Appendix C.

We conduct an experimental evaluation on two types of unsupervised transfer learning tasks, namely, self-supervised and zero-shot transfer learning. These tasks are employed to evaluate the generalizability of the proposed MSRL model under different levels of unsupervised transfer learning. To evaluate the effectiveness of the proposed MSRL method on self-supervised learning, we compare the proposed MSRL method against state-of-the-art approaches, namely, contrastive multiview clustering (CMVC) (Zhang et al., 2025), sparse multiview clustering (SparseMVC) (Liu et al., 2025), hierarchical semantic alignment and cooperative comple-

tion (HSACC) (Ding et al., 2025), bridges optimization and neural networks for efficient MVC (BONE) (Xu et al., 2026), clustering via the principle of rate reduction and pretrained models (CPP) (Chu et al., 2024), clustering via diffusion (CLUDI) (Uziel et al., 2025), masked image modeling (MIM)-refiner (Alkin et al., 2025) and TURTLE (Gadetsky et al., 2024). Additionally, CLIP zero-shot transfer (Gadetsky et al., 2024) and LaFTer (Mirza et al., 2023) are included as baselines for comparison in the zero-shot transfer learning task. Three standard metrics are employed to evaluate the clustering performance of all the competing methods, i.e., the clustering accuracy (ACC), normalized mutual information (NMI) and adjusted rand index (ARI). The source code of MSRL is publicly available at https://github.com/chenjie20/MSRL.

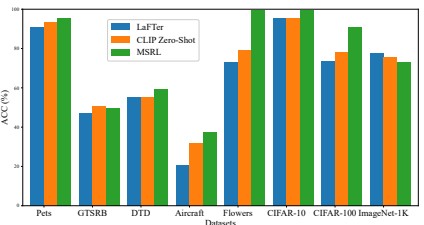

*Figure 2.* Clustering accuracy comparison between MSRL and the zero-shot transfer learning-based methods.

## 4.2. Performance Evaluation

### 4.2.1. SELF-SUPERVISED LEARNING

The experimental results of all the competing methods on the self-supervised learning task are listed in Table 1. The symbol '-' in Table 1 indicates that the competing method encounters out of memory errors when clustering is performed on a given vision dataset. The best and second-best clustering results are highlighted in bold and underlined, respectively. MSRL consistently outperforms the competing methods across all eight vision datasets. For example, it achieves ACC values of 91.00% and 73.24% on the CIFAR-100 and ImageNet-1K datasets, respectively. It significantly outperforms the other competing methods by at least 1.9% and 3.14% in terms of the ACC. Moreover, MSRL also exhibits consistent advantages across the other two evaluation metrics. In addition, TURTLE achieves competitive clustering results over the multiview clustering approaches across all vision datasets. In contrast, the multiview clustering approaches generally yield less competitive clustering results, especially for large-scale vision datasets such as CIFAR-100 and ImageNet-1K. These clustering results demonstrate the effectiveness of the proposed MSRL approach for self-supervised learning tasks.

The advantages of the proposed MSRL approach can be attributed to two key factors. First, high-level semantic knowledge is effectively transferred by leveraging multiple pretrained models, which also reduces excessive memory consumption. Second, complementary information across multiple views is explicitly exploited by enforcing consistency among assignment probability distributions, thereby mitigating the adverse effects of distribution heterogeneity across views. In contrast, traditional multiview clustering approaches typically assume homogeneous feature spaces or depend on direct feature fusion strategies, which limits their effectiveness when handling heterogeneous representations.

### 4.2.2. ZERO-SHOT TRANSFER LEARNING

We compare MSRL with zero-shot transfer learning methods that utilize descriptions of ground truth classes as a form of supervision, including CLIP zero-shot transfer (Gadetsky et al., 2024) and LaFTer (Mirza et al., 2023). The clustering accuracy comparison between MSRL and the zero-shot transfer learning-based methods is shown in Fig. 2. MSRL outperforms CLIP zero-shot transfer and LaFTer on all eight vision datasets except GTSRB and ImageNet-1K. The descriptions of the ground-truth classes provide effective external supervision for zero-shot transfer learning, which narrows the performance gap for the GTSRB and ImageNet-1K datasets. This finding can be attributed to the strong semantic alignment between class-level textual descriptions and the pretraining distribution of CLIP-based models on these datasets. In contrast, MSRL does not de-

pend on class-level semantic descriptions. Instead, it learns invariant representations from large-scale unlabeled visual data with various pretrained models in a fully unsupervised transfer manner. These invariant representations enable more flexible adaptation to diverse visual domains. Consequently, MSRL demonstrates strong robustness even in scenarios where explicit semantic supervision is available.

## 4.3. Evaluation of the Increasing Number of Pretrained Models

The clustering performance of MSRL is influenced by the number of incorporated pretrained models, since different pretrained models correspond to heterogeneous views. To investigate the effect of varying the number of views, we conduct an additional ablation study with progressively constructed model combinations. Specifically, DINOv2 ViT-g/14 is fixed as a core pretrained model, and alternative pretrained models are incrementally selected from ViT-B/32, ViT-B/16 and ViT-L/14, which results in five different combinations with increasing numbers of views. This experimental protocol enables a controlled evaluation of how clustering performance evolves as more views are incorporated. All the hyperparameters of MSRL are kept identical to those specified in the experimental settings.

Table 2 presents the clustering performance of MSRL under different combinations of pretrained models across eight vision datasets, which provides empirical validation for the incremental view analysis. When DINOv2 ViT-g/14 is combined with a single alternative pretrained model, ViT-L/14 almost emerges as the optimal choice. Moreover, the combination of DINOv2 ViT-g/14 and ViT-B/16 almost outperforms that of DINOv2 ViT-g/14 and ViT-B/32. This finding indicates that the quality of the additional view plays a critical role. When the number of pretrained models increases to three, the combination including ViT-L/14 and ViT-B/16 yields better clustering performance than the alternative combination including ViT-L/14 and ViT-B/32 does. However, when all four pretrained models are incorporated, the clustering performance is not the most competitive. Overall, these findings demonstrate that not every additional view is guaranteed to be beneficial for MSRL. In particular, lower-quality pretrained models such as ViT-B/32 may negatively affect clustering performance. Consequently, the clustering performance of MSRL can decrease as the number of views increases if low-quality pretrained models are included, which highlights the importance of both view quality and the integration of model selection.

## 4.4. Sensitivity Analysis of Batch Size During Training

Since each representation is represented as a linear combination, the batch size plays a critical role in determining the quality of the learned representations. Following the batch

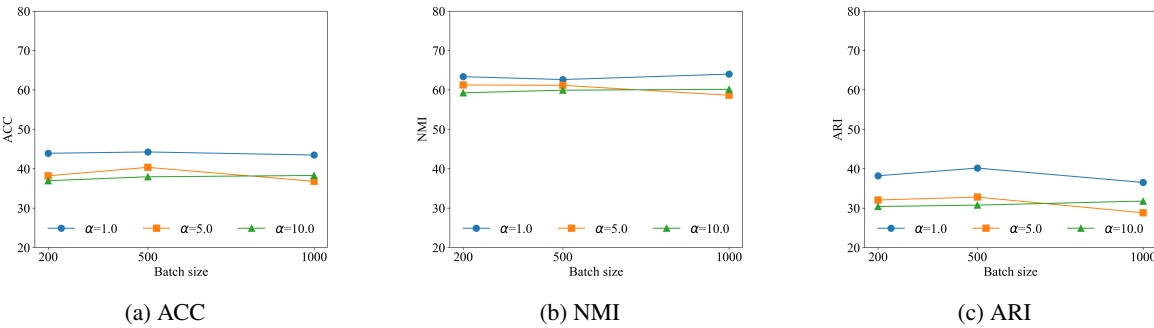

*Figure 3.* Clustering results on the GTSRB dataset under different batch sizes.

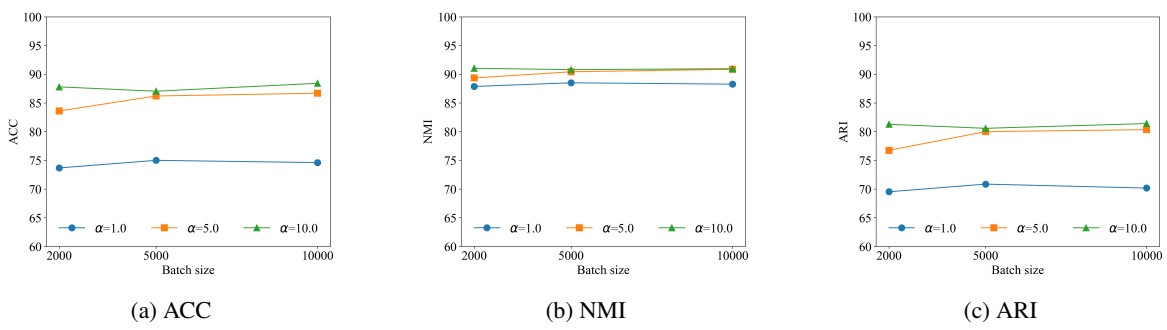

*Figure 4.* Clustering results on the CIFAR-100 dataset under different batch sizes.

size selection strategy described in the Appendix, we evaluate the effect of different batch sizes on the GTSRB and CIFAR-100 datasets, with all other experimental settings remaining unchanged. Figs. 3- 4 show clustering results on the GTSRB and CIFAR-100 datasets under different batch sizes. We observe that the clustering performance curves on both datasets exhibit only slight fluctuations in ACC, NMI, and ARI across different batch sizes. This demonstrates that MSRL is robust to the empirical choice of batch size.

### 4.5. Ablation Study

The semantic pseudolabel loss $\mathcal{L}_s$ and the cluster diversity loss $\mathcal{L}_a$ are two essential components of the overall loss in Eq. (13). In addition, the cross-view consistency loss $\mathcal{L}_c$ is incorporated to enforce consistency among assignment probability distributions across heterogeneous views, which facilitates the learning of invariant representations. To investigate the impact of the cross-view consistency loss $\mathcal{L}_c$ in Eq. (13), we conduct an ablation study for the eight vision datasets. Specifically, we construct a variant of the proposed model that omits $\mathcal{L}_c$ and employs only $\mathcal{L}_s$ and $\mathcal{L}_a$ in the overall loss, denoted as $MSRL_{avg}$. Table 3 shows the clustering performance comparison between MSRL and $MSRL_{avg}$. The ablation results show that MSRL consistently outperforms $MSRL_{avg}$ across all the evaluated datasets. This finding demonstrates the effectiveness of the cross-view consistency loss. These results indicate that $\mathcal{L}_c$ enhances assignment probability distribution consistency across mul-

tiple views, which in turn leads to significant performance improvements for MSRL. Additionally, to evaluate the contribution of FSRL, we replace the attention matrix in the aggregation operation with an identity matrix. This variant is denoted by $MSRL_v$. Table 4 shows the clustering results of $MSRL_v$ and MSRL. Unlike graph attention networks (Veličković et al., 2018), MSRL still performs well without relying on a predefined graph structure.

## 5. Conclusion

In this paper, we present the MSRL method for multiview self-representation learning across heterogeneous views. We introduce an information-passing mechanism that leverages self-representation learning to perform a feature aggregation operation on linear features, enabling information aggregation from spatially proximate neighbors. Additionally, an assignment probability distribution consistency scheme is incorporated to enforce consistency among assignment probability distributions across multiple views. This approach enables complementary information to be effectively captured in multiview self-representation learning. The proposed MSRL method learns invariant representations across different linear models. Furthermore, we provide a theoretical analysis of the assignment probability distribution consistency and incremental views. Extensive experiments demonstrate that the proposed MSRL method consistently outperforms several state-of-the-art approaches.

## Acknowledgments

This work was supported in part by National Natural Science Foundation of China (NSFC) under Grants 62576231, 62572331 and 72374089, in part by the Natural Science Foundation of Sichuan Province under Grant 2026NS-FSC0420, in part by the Open Funding of National Key Laboratory of Fundamental Algorithms and Models for Engineering Simulation, in part by Key Laboratory of Data Protection and Intelligent Management (Sichuan University), Ministry of Education under Grant SCUSACXYD202501, and in part by the Fundamental Research Funds for the Central Universities under Grant 2462025YJRC038.

## Impact Statement

Although this paper presents work whose goal is to advance the field of Machine Learning, the proposed approach relies on the representation spaces of existing foundation models. There are many potential societal consequences of our work, none of which we feel must be specifically highlighted here.

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

---

**Algorithm 1** Optimization Procedure for MSRL

---

**Input**: An unlabeled training dataset $\mathcal{D}_t = \{\mathbf{x}_1, \mathbf{x}_2, ..., \mathbf{x}_n\}$ and the unlabeled dataset $\mathcal{D}_s = \{\widehat{\mathbf{x}}_1, \widehat{\mathbf{x}}_2, ..., \widehat{\mathbf{x}}_{\widehat{n}}\}$ belonging to $C$ categories, and $L$ pretrained models.
**Parameter**: The number of iterations $epochs$, batch size $B$, parameters $\alpha > 0$ and $\beta > 0$.
**Output**: The predicted labels $\mathbf{Y} = [\widehat{y}_1, \widehat{y}_2, ..., \widehat{y}_{\widehat{n}}]$ for $\mathcal{D}_s$.

 1: **for** $t = 1$ to $epochs$ **do**
 2:    **for** each mini-batch $\mathcal{B}$ of size $B$ sampled from $\mathcal{D}_t$ **do**
 3:      **for** $l = 1$ **to** $L$ **do**
 4:         Obtain a set of features $\mathbf{H}_\mathcal{B} \in \mathbb{R}^{C \times B}$ for the mini-batch $\mathcal{B}$ via Eq. (2);
 5:         Compute the attention coefficients $a_{ij}$ for all $\mathbf{h}_i, \mathbf{h}_j \in \mathcal{B}$ via Eq. (4);
 6:         Obtain the representations of the features $\mathbf{Z}_\mathcal{B}$ in $\mathcal{B}$ via Eq. (5) by using the information-passing mechanism;
 7:         Determine the assignment probability distribution $\mathbf{s}_i$ for each unlabeled sample $\mathbf{x}_i$ produced by the $l$th component
           of the MSRL model via Eq. (6);
 8:      **end for**
 9:      Determine the cluster assignment $y_i$ for sample $\mathbf{x}_i \in \mathcal{B}$ via Eq. (8);
10:      Update the whole network by minimizing $\mathcal{L}$ in Eq. (13) over $\mathcal{B}$;
11:    **end for**
12: **end for**
13: Determine $\widehat{y}_i$ via Eq. (8) for sample $\widehat{\mathbf{x}}_i$ $(1 \leq i \leq \widehat{n})$ in $\mathcal{D}_s$.

---

# A. Implementation Details

## A.1. Information-Passing Mechanism

**Definition A.1** (Weak Neighborhood Alignment). The encoder $f$ is weakly neighborhood-aligned if, for any sample $\mathbf{x}_i$, its representation $\mathbf{z}_i$ and those of its neighbors satisfy the following condition, i.e.,

$$\mathbb{E}_{\mathbf{x}_j \in \mathbb{N}_k(\mathbf{x}_i)} \|\mathbf{z}_i - \mathbf{z}_j\|_2 \leq \varepsilon$$

where $\mathbf{x}_j \in \mathbb{N}_k(\mathbf{x}_i)$ and $\varepsilon > 0$ is a small constant.

According to Definition A.1, the difference between the representations corresponding to neighboring features approximately lies in the null space of the weight matrix $\mathbf{W}^\top$ of the shared linear layer. If row normalization is applied to $\mathbf{W}$, the following inequality holds:

$$\begin{aligned}
&\left\|\mathbf{W}^\top \left(\phi\left(\mathbf{x}_i\right) - \phi\left(\mathbf{x}_j\right)\right)\right\|_2 \\
&\leq \sqrt{d} \|\left(\phi\left(\mathbf{x}_i\right) - \phi\left(\mathbf{x}_j\right)\right)\|_2.
\end{aligned} \tag{15}$$

Row normalization mitigates undesired amplification along specific directions, making attention computation more stable. Consequently, the weak neighborhood alignment condition is more reliably satisfied in self-representation learning.

**Theorem A.2** (Local Neighborhood Alignment). *Let $\mathbf{h}_i$ and $\mathbf{h}_j$ be a neighboring feature pair, and let $\mathbf{s}_i = \delta\left(f\left(\mathbf{h}_i; \mathbf{a}_i\right)\right)$ and $\mathbf{s}_j = \delta\left(f\left(\mathbf{h}_j; \mathbf{a}_j\right)\right)$ be their corresponding assignment probability distributions, where $\delta\left(\cdot\right)$ is the softmax activation function and $f\left(\cdot; \cdot\right)$ represents an aggregation operation. The following equality holds:*

$$\mathbf{s}_i = \mathbf{s}_j$$

*if the following condition is satisfied:*

$$f\left(\mathbf{h}_i; \mathbf{a}_i\right) - f\left(\mathbf{h}_j; \mathbf{a}_j\right) = c\mathbf{1}$$

*where $c \in \mathbb{R}$ is a constant.*

*Proof.* The softmax function is translation invariant, i.e., for any vector $\mathbf{z} \in \mathbb{R}^C$,

$$\text{softmax}\left(\mathbf{z} + c\mathbf{1}\right) = \text{softmax}\left(\mathbf{z}\right).$$

Thus, it follows that

$$
\begin{aligned}
\mathbf{s}_i &= \delta\left(f\left(\mathbf{h}_i; \mathbf{a}_i\right)\right) \\
&= \delta\left(f\left(\mathbf{h}_j; \mathbf{a}_j\right) + c\mathbf{1}\right) \\
&= \delta\left(f\left(\mathbf{h}_j; \mathbf{a}_j\right)\right) \\
&= \mathbf{s}_j.
\end{aligned}
$$

$\square$

Theorem A.2 indicates that when the encoder $f$ is weakly neighborhood-aligned, neighboring features can share an identical assignment probability distribution. This result follows directly from the translation invariance of the softmax function. Furthermore, let $\mathbf{z}_i = f\left(\sigma\left(\mathbf{W}^\top; \phi\left(\mathbf{x}_i\right)\right); \mathbf{a}_i\right)$ and $\mathbf{z}_j = f\left(\sigma\left(\mathbf{W}^\top; \phi\left(\mathbf{x}_j\right)\right); \mathbf{a}_j\right)$ be two representations corresponding to a pair of neighboring samples $\mathbf{x}_i$ and $\mathbf{x}_j$, respectively. Here $\mathbf{z}_i$ and $\mathbf{z}_j$ are considered invariant representations for $\mathbf{x}_i$ and $\mathbf{x}_j$, respectively. These representations become strictly neighborhood-aligned, i.e., $\mathbf{z}_i = \mathbf{z}_j$ when the linear model satisfies $\operatorname{rank}(\mathbf{W}) = d$ and $d \leq C$. However, the opposite condition, i.e., $d > C$, often holds in high-dimensional data. This finding indicates that $\mathbf{W}$ cannot enforce strict neighborhood alignment. In contrast, weak neighborhood alignment provides more realistic and appropriate conditions for the information-passing mechanism.

### A.2. Description of Algorithm 1

An adaptive momentum-based mini-batch gradient descent method (Kingma & Ba, 2015) is used to optimize the entire network while keeping the pretrained models frozen. The entire optimization procedure of the proposed MSRL method is provided in Algorithm 1.

## B. Proofs of Main Theorems

### B.1. Proof of Theorem 3.2

*Proof.* According to the definition of $\mathbf{p}_i^{(L)}$ and the assumption, we obtain

$$
\begin{aligned}
\mathbb{E}\left[\left\|\mathbf{s}_i^{(l)} - \mathbf{p}_i^{(L)}\right\|_1\right] &= \mathbb{E}\left[\left\|\mathbf{s}_i^{(l)} - \frac{1}{L}\sum_{r=1}^{L}\mathbf{s}_i^{(r)}\right\|_1\right] \\
&\leq \frac{1}{L}\sum_{r=1}^{L}\mathbb{E}\left[\left\|\mathbf{s}_i^{(l)} - \mathbf{s}_i^{(r)}\right\|_1\right] \\
&\leq \varepsilon.
\end{aligned}
$$

As $\mathcal{H}(\cdot)$ is Lipschitz-continuous on the probability simplex $\Delta_\delta^{C-1}$, there exists a constant $Q_\delta > 0$ such that for any view $l$, we have

$$
\mathcal{H}\left(\mathbf{p}_i^{(L)}\right) - \mathcal{H}\left(\mathbf{s}_i^{(l)}\right) \leq \left|\mathcal{H}\left(\mathbf{p}_i^{(L)}\right) - \mathcal{H}\left(\mathbf{s}_i^{(l)}\right)\right| \leq Q_\delta\left\|\mathbf{p}_i^{(L)} - \mathbf{s}_i^{(l)}\right\|_1.
$$

Taking the expectation on both sides yields

$$
\mathbb{E}\left[\mathcal{H}\left(\mathbf{p}_i^{(L)}\right)\right] - \mathbb{E}\left[\mathcal{H}\left(\mathbf{s}_i^{(l)}\right)\right] \leq Q_\delta\mathbb{E}\left[\left\|\mathbf{p}_i^{(L)} - \mathbf{s}_i^{(l)}\right\|_1\right] \leq Q_\delta\varepsilon.
$$

Thus, we obtain

$$
\mathbb{E}\left[\mathcal{H}\left(\mathbf{p}_i^{(L)}\right)\right] - \frac{1}{L}\sum_{l=1}^{L}\mathbb{E}\left[\mathcal{H}\left(\mathbf{s}_i^{(l)}\right)\right] \leq Q_\delta\varepsilon.
$$

$\square$

## B.2. Introduction of Theorem B.1

**Theorem B.1** (Robbins-Monro Stochastic Approximation (Robbins & Monro, 1951)). *Let $\{u_t\}_{t\geq 1}$ be a sequence of nonnegative random variables satisfying the recursive inequality*

$$\mathbb{E}[u_{t+1}] \leq (1 - \eta_t)\,\mathbb{E}[u_t] + \eta_t \varepsilon$$

*where $\varepsilon \geq 0$ is a constant and $\{\eta_t\}_{t\geq 1}$ is a deterministic step-size sequence satisfying*

$$\eta_t > 0, \quad \sum_{t=1}^{\infty} \eta_t = \infty \quad \text{and} \quad \sum_{t=1}^{\infty} \eta_t^2 < \infty.$$

*Then, the sequence $\{\mathbb{E}[u_t]\}$ is uniformly bounded and converges to a neighborhood of size $\varepsilon$. More precisely, there exists a constant $w > 0$, depending only on the initial value $u_1$, such that*

$$\mathbb{E}[u_t] \leq w \prod_{k=1}^{t-1} (1 - \eta_k) + \varepsilon \sum_{k=1}^{t-1} \eta_k \prod_{j=k+1}^{t-1} (1 - \eta_j).$$

*In particular, when $\eta_t = 1/t$, it holds that*

$$\mathbb{E}[u_t] \leq \mathcal{O}\left(\frac{1}{t}\right) + \varepsilon.$$

## B.3. Proof of Theorem 3.4

*Proof.* A residual term is defined as

$$\mathbf{e}_i^{(L)} = \mathbf{p}_i^{(L)} - \mathbf{p}_i.$$

By the incremental updating rule in Eq. (14), we obtain

$$\mathbf{e}_i^{(L+1)} = \frac{L}{L+1}\mathbf{e}_i^{(L)} + \frac{1}{L+1}\left(\mathbf{s}_i^{(L+1)} - \mathbf{p}_i\right)$$

Taking the $\ell_1$-norm and expectation on both sides, and applying the triangle inequality, we have

$$\mathbb{E}\left[\left\|\mathbf{e}_i^{(L+1)}\right\|_1\right] \leq \frac{L}{L+1}\mathbb{E}\left[\left\|\mathbf{e}_i^{(L)}\right\|_1\right] + \frac{1}{L+1}\mathbb{E}\left[\left\|\mathbf{s}_i^{(L+1)} - \mathbf{p}_i\right\|_1\right].$$

By Assumption 3.3, we further have

$$\mathbb{E}\left[\left\|\mathbf{e}_i^{(L+1)}\right\|_1\right] \leq \frac{L}{L+1}\mathbb{E}\left[\left\|\mathbf{e}_i^{(L)}\right\|_1\right] + \frac{\varepsilon}{L+1}$$
$$= \mathbb{E}\left[\left\|\mathbf{e}_i^{(L)}\right\|_1\right] - \frac{1}{L+1}\left(\mathbb{E}\left[\left\|\mathbf{e}_i^{(L)}\right\|_1\right] - \varepsilon\right)$$

This recursive inequality corresponds to a Robbins-Monro stochastic approximation, and we get

$$\mathbb{E}\left[\left\|\mathbf{e}_i^{(L)}\right\|_1\right] \leq w\left(\frac{1}{L} + \varepsilon\right)$$

for some constant $w > 0$, which is independent of $L$. $\qquad\square$

## B.4. Proof of Theorem 3.5

*Proof.* By the incremental updating rule in Eq. (14), we have

$$\mathbf{p}_i^{(L+1)} - \mathbf{p}_i^{(L)} = \frac{1}{L+1}\left(\mathbf{s}_i^{(L+1)} - \mathbf{p}_i^{(L)}\right).$$

Thus, we obtain

$$\left\|\mathbf{p}_i^{(L+1)} - \mathbf{p}_i^{(L)}\right\|_1 = \frac{1}{L+1}\left\|\mathbf{s}_i^{(L+1)} - \mathbf{p}_i^{(L)}\right\|_1.$$

By the Lipschitz continuity of $\mathcal{H}(\cdot)$ on the probability simplex $\Delta_\delta^{C-1}$, there exists a constant $Q_\delta > 0$ such that

$$\mathcal{H}\left(\mathbf{p}_i^{(L+1)}\right) - \mathcal{H}\left(\mathbf{p}_i^{(L)}\right) \leq \left|\mathcal{H}\left(\mathbf{p}_i^{(L+1)}\right) - \mathcal{H}\left(\mathbf{p}_i^{(L)}\right)\right| \leq Q_\delta \left\|\mathbf{p}_i^{(L+1)} - \mathbf{p}_i^{(L)}\right\|_1.$$

By the triangle inequality and Assumption 3.3, we get

$$\mathbb{E}\left[\left\|\mathbf{s}_i^{(L+1)} - \mathbf{p}_i^{(L)}\right\|_1\right] \leq \mathbb{E}\left[\left\|\mathbf{s}_i^{(L+1)} - \mathbf{p}_i\right\|_1\right] + \mathbb{E}\left[\left\|\mathbf{p}_i^{(L)} - \mathbf{p}_i\right\|_1\right]$$

$$\leq \mathbb{E}\left[\left\|\mathbf{s}_i^{(L+1)} - \mathbf{p}_i\right\|_1\right] + \mathbb{E}\left[\left\|\frac{1}{L}\sum_{l=1}^{L}\left(\mathbf{s}_i^{(l)} - \mathbf{p}_i\right)\right\|_1\right]$$

$$\leq 2\varepsilon.$$

Taking the expectation on both sides and combining the above inequalities, we obtain

$$\mathbb{E}\left[\mathcal{H}\left(\mathbf{p}_i^{(L+1)}\right)\right] - \mathbb{E}\left[\mathcal{H}\left(\mathbf{p}_i^{(L)}\right)\right] \leq Q_\delta \, \mathbb{E}\left[\left\|\mathbf{p}_i^{(L+1)} - \mathbf{p}_i^{(L)}\right\|_1\right]$$

$$= \frac{Q_\delta}{L+1} \, \mathbb{E}\left[\left\|\mathbf{s}_i^{(L+1)} - \mathbf{p}_i^{(L)}\right\|_1\right]$$

$$\leq \frac{2Q_\delta \varepsilon}{L+1}.$$

□

*Table 5.* Statistics of the vision datasets.

| Name | Train | Test | Classes |
|------|-------|------|---------|
| Pets (Parkhi et al., 2012) | 3,680 | 3,669 | 37 |
| GTSRB (Stallkamp et al., 2011) | 26,640 | 12,630 | 43 |
| DTD (Cimpoi et al., 2014) | 3,760 | 1,880 | 47 |
| Aircraft (Maji et al., 2013) | 6,667 | 3,333 | 100 |
| Flowers (Nilsback & Zisserman, 2008) | 2,040 | 6,149 | 102 |
| CIFAR-10 (Krizhevsky & Hinton, 2009) | 50,000 | 10,000 | 10 |
| CIFAR-100 (Krizhevsky & Hinton, 2009) | 50,000 | 10,000 | 100 |
| ImageNet-1K (Deng et al., 2009) | 1,281,167 | 50,000 | 1,000 |

## C. Experimental Details

In this section, we conduct extensive experiments to evaluate the performance of the proposed MSRL method. All the experiments are conducted on a Linux workstation equipped with a GeForce RTX 4090 GPU (24 GB memory), an Intel(R) Xeon(R) Platinum 8336C CPU, and 128.0 GB of RAM.

### C.1. Experimental Settings

The statistics of the datasets are presented in Table 5. The source code for MSRL is implemented using the PyTorch framework (Paszke et al., 2019). The source code for all the comparison algorithms is provided by their respective authors.

#### C.1.1. PARAMETER SETTINGS

The learning rate for the proposed MSRL model was empirically set to $1 \times 10^{-3}$ for the GTSRB, DTD and CIFAR-100 datasets, and to $5 \times 10^{-4}$ for the remaining datasets. The batch size of each dataset was chosen from $\{10{,}000, 5{,}000, 1{,}000, 500, 100\}$ during training and testing. Specifically, the batch size is set to an approximate multiple of the number of clusters, chosen from $\{5, 10, 20, 50, 100\}$. Smaller multiples (e.g., 5, 10, or 20) are used for datasets with fewer clusters, while larger multiples are adopted for those with more clusters. The dropout rate was chosen from $\{0.0, 0.1, 0.2, 0.3\}$. The overall loss of the proposed MSRL model in Eq. (13) involves two hyperparameters, $\alpha$ and $\beta$. The hyperparameter $\alpha$ was selected from $\{1.0, 5.0, 10.0, 20.0\}$ via a linear search strategy, while $\beta$ was fixed at 1.0. To ensure a fair comparison, we report the best performance of all the competing methods after hyperparameter tuning.

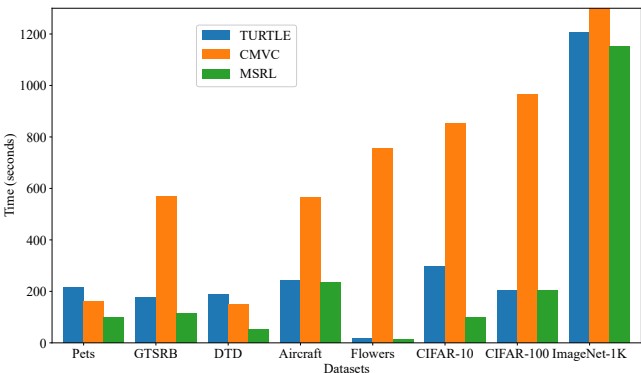

*Figure 5.* Training time comparison (in seconds) among MSRL, TURTLE and CMVC for eight vision datasets.

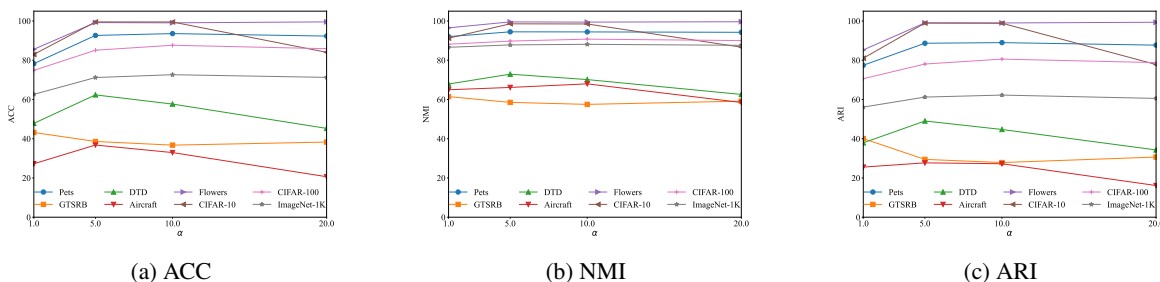

| (a) ACC | (b) NMI | (c) ARI |

*Figure 6.* Clustering results on eight vision datasets with different values of $\alpha$ in Eq. (13).

### C.2. Training Time Comparison on Self-Supervised Learning

To validate the optimization efficiency of the proposed MSRL approach, we compare the training times among MSRL, TURTLE and CMVC on all eight vision datasets. These training times are presented in Fig. 5. It can be observed that both MSRL and TURTLE significantly outperform CMVC in terms of training time, where CMVC is considered a representative method for multiview clustering. This efficiency gain can be attributed to the use of unsupervised transfer learning, which helps reduce the computational cost associated with handling raw image data. By leveraging pretrained models, both MSRL and TURTLE benefit from reduced complexity when handling high-dimensional visual features, leading to faster convergence during training. Moreover, MSRL consistently runs faster than TURTLE on most of the datasets, which can be attributed to its ability to achieve competitive performance with fewer iterations.

### C.3. Parameter Sensitivity Analysis

The semantic pseudolabel loss $\mathcal{L}_s$ and the cross-view consistency loss $\mathcal{L}_c$ in Eq. (13) are closely related to the assignment probability distributions of the samples. The parameter $\beta$ was empirically fixed at $1.0$, while the hyperparameter $\alpha$ was selected from $\{1.0, 5.0, 10.0, 20.0\}$. Since $\mathbf{s}_i^{(p)}$ and $\mathbf{s}_i^{(q)}$ in Eq. (12) tend to converge within the same probability simplex for a given sample $\mathbf{x}_i$, $\mathcal{L}_c$ in Eq. (11) acts as a local geometric regularizer that refines the cluster assignments based on pairwise linear dependencies. The objective of $\mathcal{L}_c$ is mathematically consistent with the global objective of $\mathcal{L}_s$. Consequently, we fix $\beta = 1$ as a balanced anchor for cross-view alignment. The remaining hyperparameters in the proposed MSRL method were determined according to the experimental settings. The clustering performance, including the ACC, NMI, and ARI, on the eight vision datasets under different values of $\alpha$ is shown in Fig. 6. The experimental results indicate that the proposed MSRL method achieves stable and strong performance when $\alpha$ is chosen from $\{1.0, 5.0, 10.0\}$.

### C.4. Evaluating Various Pretrained Backbone Combinations

To investigate how heterogeneous feature distributions affect the clustering performance, we evaluate combinations of three pretrained backbones: DINOv2, CLIP ViT-L/14, and ConvNeXt V2. These backbones exhibit distinct representations: transformer-based models (e.g., DINOv2 and CLIP ViT-L/14) primarily produce global semantic representations, while convolutional models (e.g., ConvNeXt V2) capture more localized spatial features. We utilize Centered Kernel Alignment

(CKA) (Kornblith et al., 2019) to measure the similarity between the feature distributions produced by different pretrained backbones. Table 6 shows the CKA values for different combinations of the pretrained backbones. The relatively low CKA scores between these backbones indicate highly disparate feature distributions.

*Table 6.* CKA values for different combinations of pretrained models.

| DINOv2 | CLIP ViT-L/14 | ConvNeXt V2 | Pets | GTSRB | DTD | Aircraft | Flowers | CIFAR-10 | CIFAR-100 | ImageNet-1K |
|---|---|---|---|---|---|---|---|---|---|---|
| ✓ | ✓ | | 0.59 | 0.45 | 0.58 | 0.47 | 0.48 | 0.53 | 0.38 | 0.39 |
| ✓ | | ✓ | 0.04 | 0.19 | 0.10 | 0.07 | 0.10 | 0.10 | 0.08 | 0.04 |
| | ✓ | ✓ | 0.05 | 0.12 | 0.05 | 0.08 | 0.21 | 0.15 | 0.09 | 0.09 |

*Table 7.* Clustering performance of MSRL with different pretrained model combinations on eight vision datasets.

| Methods | Pets | | | GTSRB | | | DTD | | | Aircraft | | | Flowers | | | CIFAR-10 | | | CIFAR-100 | | | ImageNet-1K | | |
|---|---|---|---|---|---|---|---|---|---|---|---|---|---|---|---|---|---|---|---|---|---|---|---|---|
| | ACC | NMI | ARI | ACC | NMI | ARI | ACC | NMI | ARI | ACC | NMI | ARI | ACC | NMI | ARI | ACC | NMI | ARI | ACC | NMI | ARI | ACC | NMI | ARI |
| (1) | 92.40 | 93.55 | 87.54 | 30.74 | 46.44 | 20.62 | 55.37 | 65.93 | 41.03 | 25.44 | 53.33 | 15.82 | 99.53 | 99.61 | 99.25 | 98.83 | 97.15 | 97.60 | 79.44 | 85.67 | 69.74 | 67.75 | 85.81 | 56.20 |
| (2) | 78.50 | 85.80 | 69.61 | 47.58 | 65.24 | 40.66 | 50.16 | 61.93 | 36.25 | 30.78 | 60.17 | 20.07 | 75.10 | 87.17 | 68.88 | 97.28 | 93.32 | 94.08 | 59.97 | 71.67 | 47.17 | 52.93 | 79.76 | 30.86 |
| (3) | 92.91 | 94.36 | 87.62 | 51.43 | 70.56 | 43.28 | 60.74 | 70.11 | 45.13 | 38.16 | 60.94 | 4.32 | 95.93 | 97.49 | 90.40 | 98.97 | 97.19 | 97.73 | 81.41 | 85.34 | 70.37 | 69.20 | 84.96 | 16.50 |
| MSRL | **95.83** | **95.83** | **92.14** | **49.78** | **65.89** | **46.65** | **62.34** | **72.89** | **49.03** | **37.65** | **66.96** | **27.72** | **99.71** | **99.76** | **99.58** | **99.57** | **98.74** | **99.05** | **91.00** | **92.34** | **84.72** | **73.24** | **88.26** | **62.80** |

*Table 8.* Comparison of the performance between MSRL and the standard LR baselines.

| Methods | Pets | GTSRB | DTD | Aircraft | Flowers | CIFAR-10 | CIFAR-100 | ImageNet-1K |
|---|---|---|---|---|---|---|---|---|
| LR+DINOv2 | 94.99 | **93.25** | 81.49 | 68.23 | 99.04 | 97.89 | 85.98 | **86.32** |
| LR+CLIP ViT-L/14 | **96.46** | 80.10 | **84.79** | **87.79** | **99.69** | 99.40 | **93.43** | 84.43 |
| MSRL | 95.83 | 49.78 | 62.34 | 37.65 | **99.71** | **99.57** | 91.00 | 73.24 |

We consider three backbone combinations: (1) DINOv2 + ConvNeXt V2; (2) CLIP ViT-L/14 + ConvNeXt V2; and (3) DINOv2 + CLIP ViT-L/14 + ConvNeXt V2. The experimental results of these combinations on all of the datasets are reported in Table 7. For smaller-scale datasets, such as Pets, GTSRB, and DTD, the combination of all three heterogeneous backbones consistently outperforms combinations involving only two backbones. However, on larger-scale datasets such as CIFAR-100 and ImageNet-1K, the additional performance gains become marginal or inconsistent. Moreover, combining three heterogeneous backbones incurs higher computational cost as the number of features increases. Finally, the significant distributional discrepancy between DINOv2 (or CLIP ViT-L/14) and ConvNeXt V2 leads to degraded clustering performance for these combinations when compared with MSRL.

## C.5. Visualizations

To empirically validate the effectiveness of the view-invariant representations obtained by MSRL, we employ t-SNE to visualize the feature distributions (van der Maaten & Hinton, 2008) on the Pets, DTD, and Aircraft datasets. Specifically, we extract and compare features from three distinct stages of the MSRL model: the original heterogeneous features derived from the two frozen pretrained backbones, the linear features obtained after the view-specific linear layers, and the final aggregation features generated by the information-passing mechanism via the self-attention layers.

Figs. 7- 12 show the t-SNE visualizations across the Pets, DTD, and Aircraft datasets. This hierarchical visualization shows how the feature spaces gradually change from separate pretraining distributions to a shared representation space. The visualization results also demonstrate two key advantages of the proposed MSRL method. First, in terms of separability, the final aggregation features have more compact intra-class clusters and clearer inter-class boundaries than both the original and linear features. This indicates a better clustering structure. Second, regarding invariance, the aggregation features from different views of the same category cluster more consistently and clearly. This finding confirms that the proposed assignment probability distribution consistency scheme effectively reduces the distribution gap across heterogeneous views. Consequently, these results show that MSRL successfully learns view-invariant representations and improves the overall clustering structure.

## C.6. Supervised Linear Probing

We employ a standard logistic regression (LR) model (Jr et al., 2013) to evaluate the original features generated by the two pretrained backbones, i.e., DINOv2 and CLIP ViT-L/14, in a supervised manner. The experimental results are reported in Table 8. These results demonstrate that MSRL achieves competitive performance on the Pets dataset and outperforms all of the supervised baselines on the Flowers and CIFAR-10 datasets.

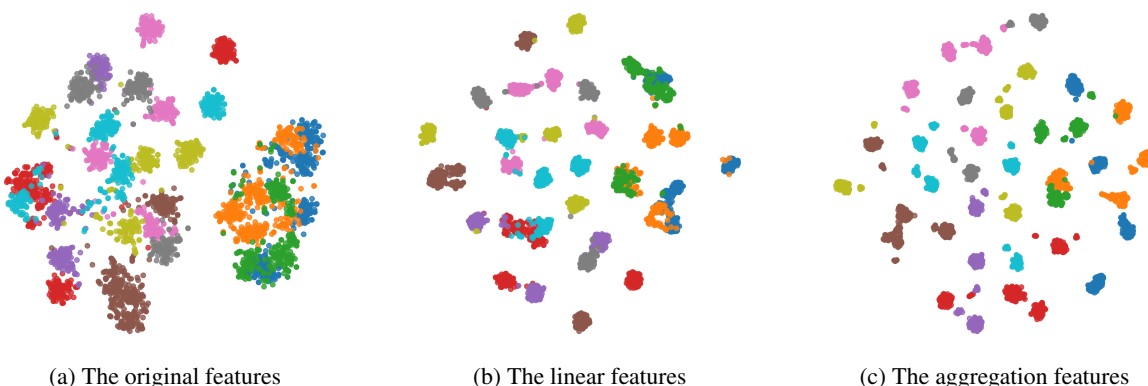

|             (a) The original features             |             (b) The linear features             |             (c) The aggregation features             |

*Figure 7.* The t-SNE visualization shows three levels of features produced by MSRL on the Pets dataset, where the original features for View 1 are extracted by using CLIP ViT-L/14.

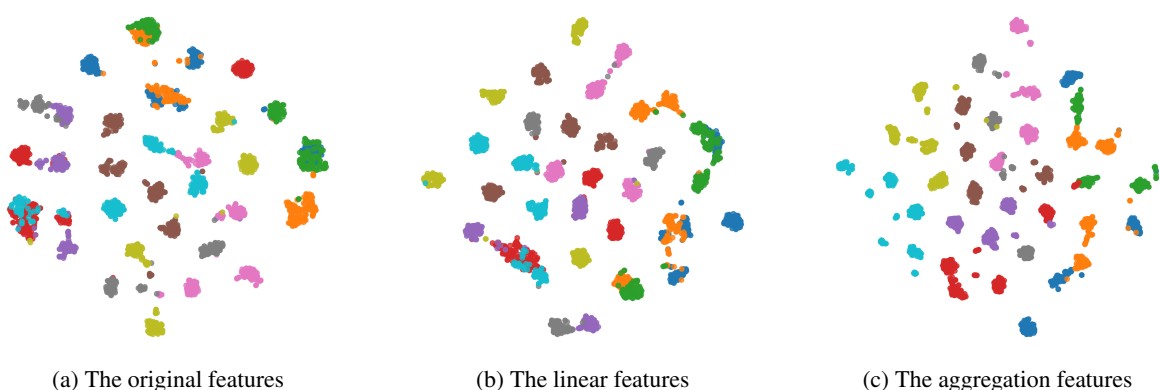

|             (a) The original features             |             (b) The linear features             |             (c) The aggregation features             |

*Figure 8.* The t-SNE visualization shows three levels of features produced by MSRL on the Pets dataset, where the original features for View 2 are extracted by using DINOv2.

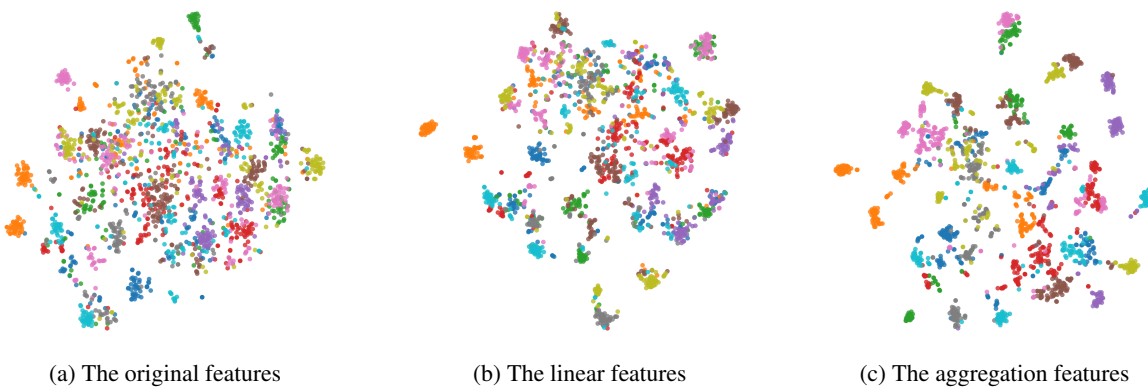

|             (a) The original features             |             (b) The linear features             |             (c) The aggregation features             |

*Figure 9.* The t-SNE visualization shows three levels of features produced by MSRL on the DTD dataset, where the original features for View 1 are extracted by using CLIP ViT-L/14.

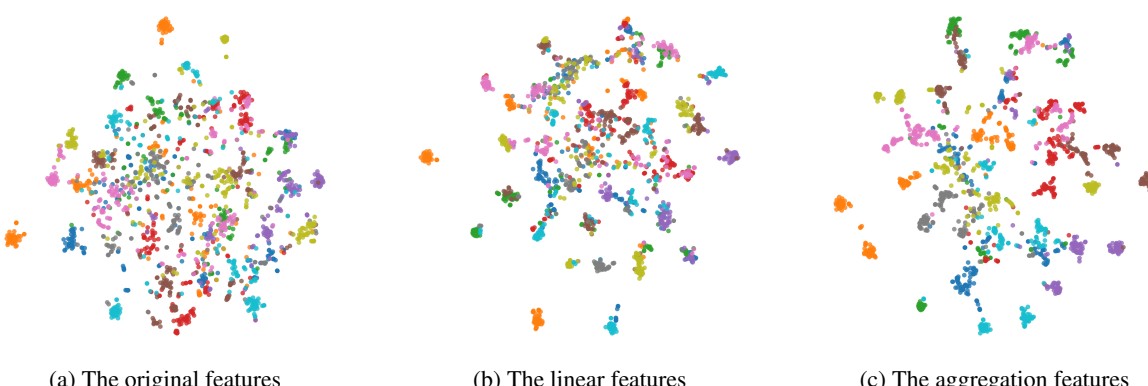

(a) The original features     (b) The linear features     (c) The aggregation features

*Figure 10.* The t-SNE visualization shows three levels of features produced by MSRL on the DTD dataset, where the original features for View 2 are extracted by using DINOv2.

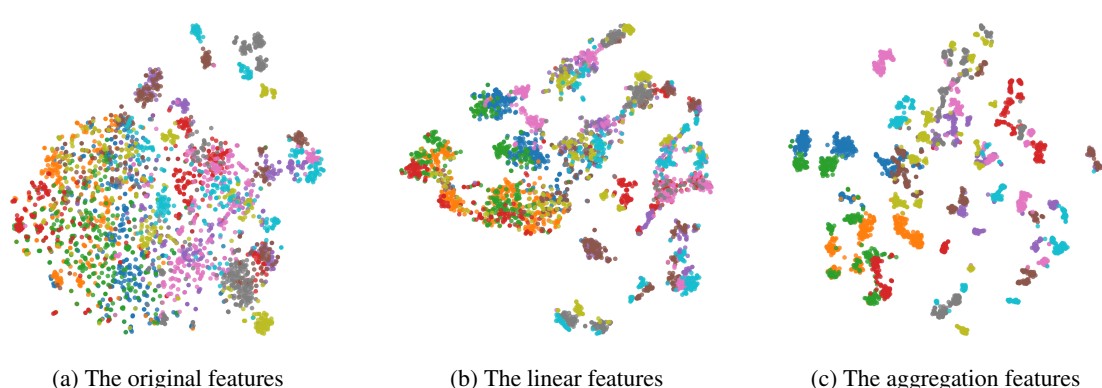

(a) The original features     (b) The linear features     (c) The aggregation features

*Figure 11.* The t-SNE visualization shows three levels of features produced by MSRL on the Aircraft dataset, where the original features for View 1 are extracted by using CLIP ViT-L/14.

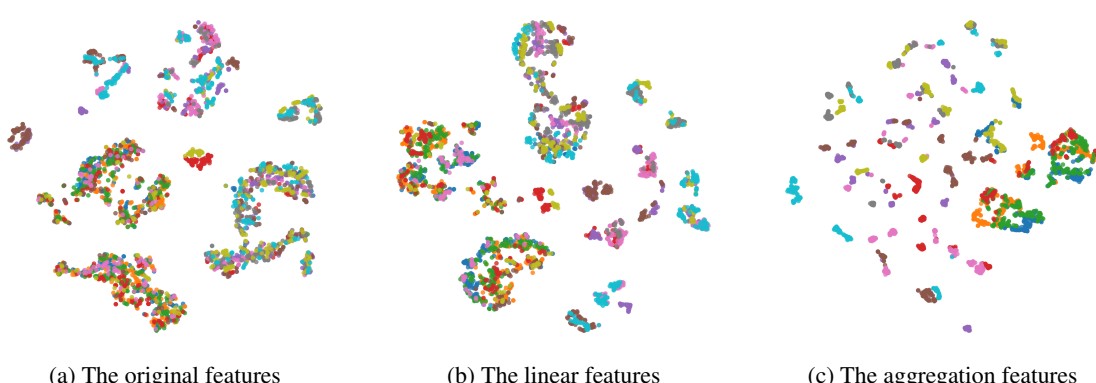

(a) The original features     (b) The linear features     (c) The aggregation features

*Figure 12.* The t-SNE visualization shows three levels of features produced by MSRL on the Aircraft dataset, where the original features for View 2 are extracted by using DINOv2.

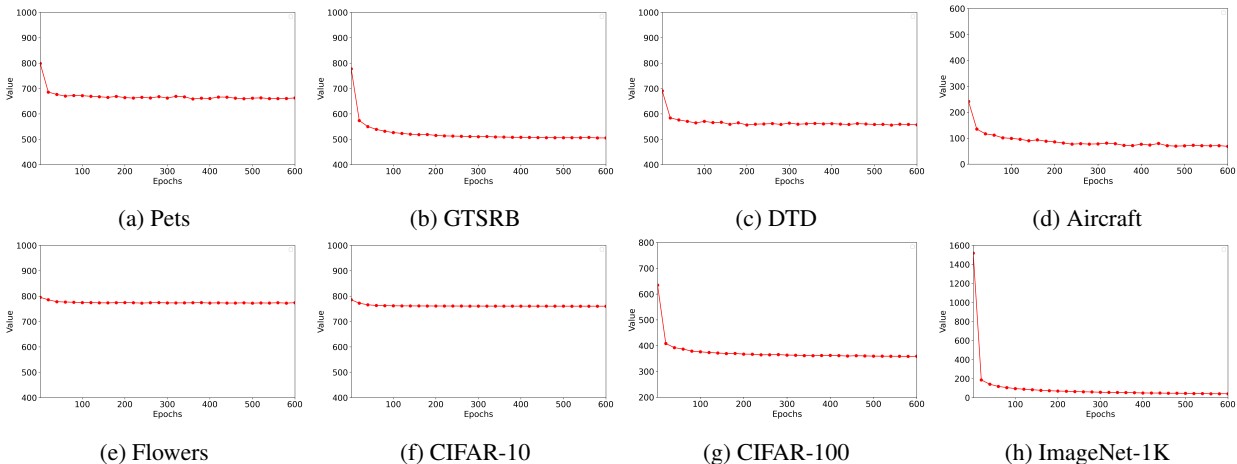

*Figure 13.* Convergence results of MSRL on all of the datasets.

## C.7. Convergence Analysis

We empirically investigate the convergence properties of the proposed MSRL model across all eight vision datasets. Fig. 13 shows the convergence curves of the overall loss function in Eq. (13). We observe that the loss values typically decrease sharply within the first few dozen iterations and then remain stable or exhibit slight fluctuations throughout the remaining training process. This finding demonstrates that MSRL achieves strong empirical convergence in practice.

