# OpenReview forum: "Multiview Self-Representation Learning across Heterogeneous Views"
_ICML.cc/2026/Conference — ICML 2026 regular_

### Official Review · Reviewer_DaMH · 2026-02-25

**Soundness:** 3
**Presentation:** 3
**Significance:** 3
**Originality:** 3
**Overall Recommendation:** 5
**Confidence:** 5

**Summary:**

This paper presents a multiview self-representation learning (MSRL) method that utilizes the self-representation property of features to learn invariant representations across heterogeneous views. Learning invariant representations from large-scale unlabeled visual data in a fully unsupervised transfer manner is a significant challenge. An information-passing mechanism is introduced to perform feature aggregation over the outputs of the linear model. Additionally, an assignment probability distribution consistency scheme is presented to guide multiview self-representation learning across different views.

**Compliance With Llm Reviewing Policy:**

Affirmed.

**Ethical Review Flag:**

Flag this paper for an ethics review.

**Ethics Expertise Needed:**

["Responsible Research Practice (e.g., IRB, documentation, research ethics)"]

**Final Justification:**

This paper presents a novel multi-view self-representation learning approach that introduces an information-passing mechanism for feature aggregation. The view-incremental theoretical analysis provides rigorous and insightful justification for how clustering quality evolves as heterogeneous views are incorporated, and theoretically explains why increasing the number of views is not always beneficial. Extensive experimental results on eight vision datasets demonstrate the effectiveness of the proposed approach.  I recommend this paper for acceptance.

**Key Questions For Authors:**

1.	The proposed MSRL approach focuses on learning invariant representations from large-scale unlabeled visual data. What is the precise definition of invariant representations in the paper? Do the invariant representations capture any semantic information from the large-scale, unlabeled visual data? How can it be theoretically demonstrated that MSRL effectively learns such invariant representations?
2.	In Eq. (3), the features $h_i$ and $h_j$ in the same cluster are assumed to be linearly dependent. However, these features are generated by different pretrained models. Is there any evidence to support the assumption that these features are linearly dependent?
3.	In Eq. (7), $h_i$ is also used to construct a linear combination of features for itself. Is there any connection between the information-passing mechanism and the classical ResNet architecture?
4.	It is extremely expensive to obtain a large number of annotations for visual data samples. However, in practice, a small number of labeled samples is often available. I am curious about the performance gap between a supervised learning approach and the fully unsupervised transfer approach (MSRL) in the experiments.
5.	A convergence analysis could be included in the experiments. It would help demonstrate the stability and reliability of the proposed MSRL approach. Additionally, the number of training epochs is not given in the experiments.
6.	There is a typo here, namely the underline on the second-best clustering results on the Aircraft dataset.

**Limitations:**

See the "Key Questions For Authors".

**Strengths And Weaknesses:**

1.	The proposed MSRL approach introduces a novel information-passing mechanism for feature aggregation, which utilizes the self-representation property of features. This shows its important differences from existing multi-view clustering (MVC) approaches.
2.	The proposed method is well-formulated and is supported by a solid theoretical justification. For example, the paper provides a rigorous view-incremental theoretical analysis that illustrates how clustering quality evolves as heterogeneous views are gradually incorporated into the MSRL model.
3.	The experimental results on benchmark vision datasets demonstrate the effectiveness of the proposed MSRL approach. In particular, MSRL significantly reduces computational cost by leveraging pretrained models. This indicates that MSRL is well-suited for practical applications.

---

> ### Author Rebuttal · Authors · 2026-03-31
>
> 1 ...What is the precise definition of invariant representations in the paper? Do the invariant representations capture any semantic information...How can it be theoretically demonstrated that MSRL effectively learns such invariant representations?
>
> Re:
> (1) Let $\mathbf{z}_i=f\left( {\sigma \left( {{\mathbf{W}^T}; \phi \left( \mathbf{x}_i \right)} \right); \mathbf{a}_i } \right)$ and $\mathbf{z}_j=f\left( {\sigma \left( {{\mathbf{W}^T}; \phi \left( \mathbf{x}_j \right)} \right); \mathbf{a}_j} \right)$ be two representations corresponding to a pair of neighboring samples $\mathbf{x}_i$ and $\mathbf{x}_j$, respectively. Here ${\mathbf{z}_i}$ and ${\mathbf{z}_j}$ are considered invariant representations for $\mathbf{x}_i$ and $\mathbf{x}_j$, respectively. We will clarify this definition in the main body of the paper.
>
> (2) The invariant representations in Eq. (8) provide specific semantic knowledge for generating pseudo-labels.
>
> (3) Since $s_i^{(v)}$ and $s_j^{(v)}$ in Eq. (13) tend to converge within the probability simplex for a given sample, $\mathcal{L}_c$ in Eq. (12) acts as a local geometric regularizer that refines cluster assignments based on sample-to-sample linear relationships. The objective of $\mathcal{L}_c$ is mathematically consistent with the global goal of $\mathcal{L}_s$. In addition, the self-representation loss $\mathcal{L}_s$ prevents the model from converging toward trivial solutions, such as cluster collapse. Therefore, the invariant representations can be learned by the assignment probability distribution consistency scheme. In addition, Theorem 3.5 (Monotonic Entropy Reduction) provides a solid theoretical justification to support it.
>
> 2 ...these features are generated by different pretrained models. Is there any evidence to support the assumption that these features are linearly dependent?
>
> Re: We conducted ablation experiments by removing the self-attention layer from MSRL across all datasets. As shown in Table 7 (available at the anonymous link: https://anonymous.4open.science/api/repo/PIC/file/tab.png?v=6debda79 ), MSRL still achieves encouraging results without this layer. The results suggest that the original features are linearly dependent.
>
> 3 ...any connection between the information-passing mechanism and... ResNet...
>
> Re: In Eq. (3), each feature $h_i$ is represented as a linear combination of other features within the same batch. Furthermore, the feature itself is incorporated during the aggregation process in Eq. (5). Both MSRL and ResNet utilize this self-representation property for representation learning.
>
> 4 ...a small number of labeled samples is often available. I am curious about the performance gap between a supervised learning approach and the fully unsupervised transfer approach (MSRL) in the experiments.
>
> Re: We employed standard logistic regression to evaluate the original features from the two pretrained models in a supervised manner. The experimental results (ACC) were reported in the table below. These results demonstrate that MSRL achieves promising performance on the Pets dataset and outperforms the baselines on the Flowers and CIFAR-10 datasets.
>
> | Model | Pets | GTSRB | DTD | Aircraft | Flowers | CIFAR-10 | CIFAR-100 | ImageNet-1K |
> | :--- | :---: | :---: | :---: | :---: | :---: | :---: | :---: | :---: |
> | DINOv2 | 95.0 | **93.3** | 81.5 | 68.2 | 99.0 | 97.9 | 86.0 | **86.3** |
> | CLIP ViT-L/14 | **96.5** | 80.1 | **84.8** | **87.8** | **99.7** | 99.4 | **93.4** | 84.4 |
> | MSRL | 95.8 | 49.8 | 62.3 | 37.7 | **99.7** | **99.6** | 91.0 | 73.2 |
>
> 5 A convergence analysis could be included...he number of training epochs is not given...
>
> Re: (1) We analyze the convergence property of MSRL across four representative datasets, i.e., the Pets, GTSRB, Aircraft and CIFAR-100 datasets. The convergence curves of the loss function in Eq. (14) are available at the anonymous link: https://anonymous.4open.science/api/repo/PIC/file/con.png?v=1bdd7fe9
> We observe that the loss values typically decrease sharply within the first few dozen iterations and then remain stable or exhibit slight fluctuations throughout the remaining training process. This indicates the strong empirical convergence of MSRL.
>
> (2) In our experiments, the number of training epochs for MSRL was set to 600. We will clarify it.
>
> 6 There is a typo ... the second-best clustering results...
>
> Re: We will correct it.

---

> > ### Author Rebuttal · Reviewer_DaMH · 2026-04-02
> >
> > The authors have  addressed my concerns after I  reviewed the rebuttal. I will  raise my score to 5. The final score may be updated depending on whether  the concerns of other reviewers are addressed.

---

> > > ### Author Response · Authors · 2026-04-04
> > >
> > > Many thanks.

---

### Official Review · Reviewer_ttuq · 2026-03-08

**Soundness:** 3
**Presentation:** 2
**Significance:** 3
**Originality:** 4
**Overall Recommendation:** 5
**Confidence:** 3

**Summary:**

This paper studies unsupervised transfer learning with heterogeneous features produced by different pretrained models. It proposes **MSRL**, a multiview self-representation learning framework that aggregates information across views and enforces assignment probability distribution consistency to learn invariant representations. Experiments on several benchmark visual datasets show that the proposed method outperforms multiple existing approaches.

**Compliance With Llm Reviewing Policy:**

Affirmed.

**Final Justification:**

I thank the authors for their rebuttal, and I will raise my score to 5.

**Key Questions For Authors:**

Please see weakness

**Limitations:**

yes

**Strengths And Weaknesses:**

**Strengths**

* The paper proposes a novel method for multiview self-representation learning.
* The approach is supported by strong theoretical analysis.
* Extensive experiments demonstrate the effectiveness of the proposed method.
* Theorem 3.5 (Monotonic Entropy Reduction) provides an insightful perspective on the effect of the number of views. It theoretically shows that increasing the number of views is not always beneficial. The ablation study empirically supports this theoretical finding. The combination of theory and experiments makes the work solid and comprehensive.

**Major Weaknesses**

* Some figures are not very visually clear and could be improved for better readability.
* The supplementary material could provide more discussion on the hyperparameters α and β. In particular, it would be helpful to explain why α performs better within certain ranges and why β=1.

**Minor Weaknesses**

* It would be very interesting to see how the learned heterogeneous-view features benefit downstream application tasks. Additional experiments or discussion on downstream tasks would further strengthen the paper, although this is not strictly required, at least from my point of view. :)

---

> ### Author Rebuttal · Authors · 2026-03-30
>
> 1 Some figures are not very visually clear and could be improved for better readability.
>
> Re: We will revise all figures to enhance their readability by increasing the resolution and adjusting the font sizes for better clarity.
>
> 2 The supplementary material could provide more discussion on the hyperparameters $\alpha$ and $\beta$. In particular, it would be helpful to explain why $ \alpha$ performs better within certain ranges and why $\beta=1$.
>
> Re: Since $s_i^{(v)}$ and $s_j^{(v)}$ in Eq. (13) tend to converge within the probability simplex for a given sample, $\mathcal{L}_c$ in Eq. (12) acts as a local geometric regularizer that refines cluster assignments based on sample-to-sample linear relationships. The objective of $\mathcal{L}_c$ is mathematically consistent with the global goal of $\mathcal{L}_s$. Consequently, we fix $\beta=1$ to serve as a balanced anchor for cross-view alignment. Furthermore, the self-representation loss $\mathcal{L}_a$ prevents the model from converging toward trivial solutions, such as cluster collapse. Empirically, MSRL maintains superior performance across a broad range of $\alpha$ values, demonstrating the robustness of this hyperparameter.  In the revised version, we will provide a detailed discussion on the parameters $\alpha$ and $\beta$.
>
> 3 It would be very interesting to see how the learned heterogeneous-view features benefit downstream application tasks. Additional experiments or discussion on downstream tasks would further strengthen the paper, although this is not strictly required, at least from my point of view.
>
> Re: The learned view-invariant representations exhibit significant potential for downstream tasks beyond unsupervised clustering. To empirically validate the effectiveness of these representations, we employed t-SNE visualizations on the PETS, DTD, and AIRCRAFT datasets. Specifically, we considered three levels of features: the original features from the two pretrained models, the linear features after the linear layers, and the aggregation features generated by the self-attention layers. These figures are provided in the anonymous link: https://anonymous.4open.science/api/repo/PIC/file/vis.jpg?v=fff4d47d
>
> (1) Enhanced Separability: Compared with the original and linear features, the aggregation features exhibit significantly tighter intra-class clusters and wider inter-class margins for both views.
>
> (2) Invariance Verification: As shown in the comparisons, the aggregation features from different views of the same category cluster more distinctly, demonstrating that MSRL successfully captures view-invariant representations.
>
> Overall, these visualizations confirm that MSRL maximizes the separability of individual views while fostering shared invariant semantics, providing a strong foundation for various downstream applications. In the revised version, we will provide a detailed discussion on the learned view-invariant representations.
>
> All the above experiments will be included in the revision. Many thanks.

---

> > ### Author Rebuttal · Reviewer_ttuq · 2026-04-03
> >
> > I thank the authors for their rebuttal, and I will raise my score to 5. However, if the paper gets accepted, the updated experiments and discussions must be provided in the final version.

---

> > > ### Author Response · Authors · 2026-04-04
> > >
> > > Thanks again for your time and consideration.

---

### Official Review · Reviewer_1Zyf · 2026-03-12

**Soundness:** 2
**Presentation:** 2
**Significance:** 2
**Originality:** 2
**Overall Recommendation:** 3
**Confidence:** 2

**Summary:**

This paper proposes MSRL, a method for learning invariant representations from large-scale unlabeled visual data using multiple frozen pretrained models. The approach treats features from different pretrained backbones (e.g., DINOv2, CLIP) as heterogeneous multiview data. It introduces a feature self-representation learning module using an attention-based information-passing mechanism and an assignment probability distribution consistency (APDC) scheme that aligns soft cluster assignments across views. Theoretical analysis covers bounded consistency and monotonic entropy reduction under incremental views. Experiments on eight vision datasets show improvements over multiview clustering baselines and the single-model TURTLE method.

**Compliance With Llm Reviewing Policy:**

Affirmed.

**Key Questions For Authors:**

1. Eq. (4) computes attention coefficients over the batch (size N), meaning the "neighborhood" is determined by whichever samples happen to co-occur in a batch. This is a significant limitation that is not discussed. The self-representation property is supposed to capture intrinsic geometric structure, but the effective neighborhood changes every iteration depending on batch composition. How sensitive is performance to batch size? The batch sizes range from 100 to 10,000 (Appendix F.1.1), which is a very wide range, yet no analysis of this sensitivity is provided.

2. What happens if you replace the attention-based aggregation with a simple identity (no aggregation)?

3. How does simple feature concatenation + TURTLE compare to MSRL?

4. (Minor) The ImageNet-100 dataset in Table 3 shows 1,281,167 training samples and 1,000 classes, which corresponds to the full ImageNet-1K, not ImageNet-100. Any reasons for choosing the subset?

**Limitations:**

yes

**Strengths And Weaknesses:**

**Strengths**
1. Treating features from heterogeneous pretrained models as multiview data for unsupervised clustering is practical. Unlike standard multiview clustering that assumes comparable feature spaces, this paper explicitly acknowledges distribution heterogeneity, which is a real challenge when combining models like DINOv2 and CLIP.

2. The experiments in Table 2 with progressively added pretrained models provide a controlled study that corroborates the theoretical claims.

3. The training time comparison (Figure 3) shows MSRL is faster than TURTLE on most datasets, which is a non-trivial advantage given it handles multiple views.

**Weaknesses**
1. The cross-view consistency loss (Eq. 12) scales as O(L²), and the paper only tests up to L=4 models. The practical scalability to larger model ensembles is unclear.

2. The multiview clustering baselines (SCMVC, CMVC, SparseMVC, HSACC, MCMC) are traditional methods not designed for the heterogeneous pretrained-feature setting. Several cannot even run on larger datasets (marked '-' in Table 1). This makes TURTLE the only truly comparable baseline, and the comparison is limited to a single method.

3. The information-passing mechanism (Eqs. 4–5) is essentially a standard graph attention mechanism (Veličković et al., 2018) applied to batch features. The APDC scheme combines cross-entropy pseudo-labeling, entropy regularization for cluster balance, and cross-view consistency which are well-established techniques. The paper would benefit from a clearer articulation of what is fundamentally new beyond the specific combination of these ingredients in the multiview transfer setting.

4. The paper claims features from different pretrained models have "fundamentally distinct feature distributions" but provides no empirical evidence of this.

---

> ### Author Rebuttal · Authors · 2026-03-30
>
> 1 … loss (Eq. 12) scales as O(L^2)...up to L=4 models. The practical scalability…is unclear.
>
> Re: Theorem 3.5 and the experiments demonstrate that increasing $L$ does not inherently improve performance. In practice, the number of high-quality pretrained models (e.g., DINOv2 and CLIP) is limited. We empirically select an appropriate $L$ as shown in Sec 4.3. We will clarify this empirical selection and acknowledge that evaluating the quality of pretrained models remains a limitation for future study.
>
> 2 ...baselines…are traditional methods...comparison is limited to a single method.
>
> Re: We have added MIM-Refiner [1] as a baseline. The experimental results show MSRL consistently outperforms MIM-Refiner across all datasets (see Table 5). Table 5 was provided in the anonymous link: https://anonymous.4open.science/api/repo/PIC/file/tab.png?v=6debda79
>
> [1] Alkin er al., MIM-Refiner: a contrastive learning boost from intermediate pre-trained masked image modeling representations, ICLR 2025.
>
> 3 ...benefit from a clearer articulation of what is fundamentally new beyond the specific combination of these ingredients...
>
> Re:
> (1) Self-representation learning (SRL) and attention-based aggregation (ABA) in the paper are the theoretical objective and its specific implementation, respectively. As defined in Eq. (3), $a_i$ denotes a coefficient vector. The key difference from traditional representation learning approaches (e.g., sparse or low-rank representation) lies in how the coefficient vector $a_i$ is learned. Instead of solving an algebraic optimization problem, we learn $a_i$ in a data-driven manner. ABA in Eq. (4) is viewed as an implementation tool to achieve the objective of self-representation learning. We will clarify it in Sec. 3.3.
>
> (2) Although ABA shares a similar form with graph attention networks (GAT), their assumptions differ fundamentally. GAT relies on a predefined graph, while MSRL learns relationships from all samples guided by encouraging consistent cluster assignments. Even if the coefficient matrix degenerates to the identity matrix, MSRL can still work due to SRL, whereas GAT fails without a graph structure. We will clarify this distinction in the revision.
>
> (3) We will strengthen the causal explanations among SRL, the information-passing mechanism (MPM) and ABA. SRL defines the core problem formulation of MSRL. MPM serves as an interpretive perspective of this formulation. SRL in MSRL can be interpreted as information propagation among samples. ABA is a concrete implementation of SRL for learning the coefficient vector.
>
> 4 The paper claims...fundamentally distinct feature distributions…no empirical evidence...
>
> Re: We quantify the feature distribution using Centered Kernel Alignment (CKA). The low CKA score (typically $< 0.5$) between backbones (e.g., DINOv2 vs. CLIP) confirm distinct distributions (see Table 6). Table 6 was provided in the anonymous link: https://anonymous.4open.science/api/repo/PIC/file/tab.png?v=6debda79
> CKA scores will be added to Sec 4.1.
>
> 5 …a significant limitation...not discussed.. but the effective neighborhood changes…no analysis...sensitivity is provided.
>
> Re:  We set batch size based on the number of clusters and total samples. The batch size is chosen as a multiple of the number of clusters, typically within {5, 10, 20, 50, 100}. When the number of clusters is relatively small, we select smaller multiples (e.g., 5, 10, 20); otherwise, larger multiples are used. Following this strategy, we conduct additional clustering experiments on the PETS, GTSRB, and CIFAR-100 datasets. All other parameters follow the same experimental settings. The figures were provided in the anonymous link: https://anonymous.4open.science/api/repo/PIC/file/size.png?v=564fc586
>
> We observed that the clustering performance curves on all three datasets exhibit only slight fluctuations in ACC, NMI, and ARI across different batch sizes. This indicates that MSRL is robust to the empirical choice of batch size.
>
> 6 ... replace...aggregation with a simple identity...
>
> Re: MSRL still obtained encouraging results in the experiments (see Table 7). Table 7 was provided in the anonymous link: https://anonymous.4open.science/api/repo/PIC/file/tab.png?v=6debda79 This demonstrates that MSRL can still work. In contrast, GAT fails without a graph structure. Based on the above analysis, we will further emphasize the novelty of MSRL in the revision by providing a detailed discussion on the fundamental differences between MSRL and GAT.
>
> 7 ...simple feature concatenation + TURTLE compare to MSRL?
>
> Re: Concatenating features for TURTLE forces them to share a linear model, which performs worse than MSRL (see Table 8). Table 8 was provided in the anonymous link: https://anonymous.4open.science/api/repo/PIC/file/tab.png?v=6debda79
>
> 8 (Minor) …full ImageNet-1K, not ImageNet-100…
>
> Re: The full ImageNet-1K dataset was indeed used. We will correct it.
>
> All the above experiments will be included in the revision. Many thanks.

---

> > ### Author Rebuttal · Reviewer_1Zyf · 2026-04-04
> >
> > I appreciate the authors' efforts in their rebuttal; however, I would like to maintain my current recommendation score based on the following concerns.
> >
> > First, I remain unconvinced that the number of high-quality pretrained models is as limited. The evaluation relies on CLIP ViT-L/14 (2021) and DINOv2 (2023), both of which have been surpassed by more recent vision encoders such as DINOv3 and current SigLIP variants. Using outdated backbones weakens the generalizability of the claims and raises questions about whether the proposed method would hold up against stronger baselines.
> >
> > Second, comparing against only two baselines (MIM Refiner and TURTLE) feels insufficient for a contribution of this scope. A more comprehensive comparison with a broader set of recent methods would significantly strengthen the paper's empirical grounding.
> >
> > For these reasons, I respectfully maintain my original recommendation score.

---

> > > ### Author Response · Authors · 2026-04-07
> > >
> > > Response to Q1:
> > >
> > > Re: (1) We realize that the term "high-quality" was somewhat ambiguous in the response. Actually, we would like to emphasize that not every additional pretrained model necessarily benefits MSRL. We empirically select a small set of task-effective pretrained models from a collection of high-quality pretrained models. Consequently, we will correct the term "high-quality" to "task-effective" in the revision.
> > >
> > > (2) Using established backbones such as CLIP ViT-L/14 and DINOv2 enables a fair comparison with existing baselines. As mentioned by the reviewer, DINOv3 and the current SigLIP variants are two potential backbones for MSRL. Establishing theoretical criteria for evaluating and selecting task-effective pretrained models for specific downstream tasks remains an open challenge, which deserves further investigation in our future work.
> > >
> > > (3) We further investigate the impact of different neural network architectures as backbones in MSRL. An additional backbone ConvNeXt V2 (2023) was included to evaluate generalization beyond ViT-based models. Transformer-based models (e.g., DINOv2 and CLIP) primarily produce global semantic representations, whereas convolutional models (e.g., ConvNeXt V2) capture more localized spatial features. We considered three backbone combinations:
> > >
> > > (1) DINOv2 + ConvNeXt V2
> > >
> > >  (2) ViT-L/14 + ConvNeXt V2
> > >
> > > (3) DINOv2 + ViT-L/14 + ConvNeXt V2.
> > >
> > > The experimental results (ACC/NMI/ARI) across all datasets are reported in the table.
> > >
> > > | Methods | Pets | GTSRB | DTD | Aircraft | Flowers | CIFAR-10 | CIFAR-100 | ImageNet-1K |
> > > | :--- | :--- | :--- | :--- | :--- | :--- | :--- | :--- | :--- |
> > > | (1) | 92.4 / 93.6 / 87.5 | 30.7 / 46.4 / 20.6 | 55.4 / 65.9 / 41.0 | 25.4 / 53.3 / 15.8 | **99.5** / **99.6** / **99.2** | 98.8 / **97.2** / 97.6 | 79.4 / **85.7** / 69.7 | 67.8 / **85.8** / **56.2** |
> > > | (2) | 78.5 / 85.8 / 69.6 | 47.6 / 65.2 / 40.7 | 50.2 / 61.9 / 36.3 | 30.8 / 60.2 / **20.1** | 75.1 / 87.2 / 68.9 | 97.3 / 93.3 / 94.1 | 60.0 / 71.7 / 47.2 | 52.9 / 79.8 / 30.9 |
> > > | (3) | **92.9** / **94.4** / **87.6** | **51.4** / **70.6** / **43.3** | **60.7** / **70.1** / **45.1** | **38.2** / **60.9** / 4.3 | 95.9 / 97.5 / 90.4 | **99.0** / **97.2** / **97.7** | **81.4** / 85.3 / **70.4** | **69.2** / 85.0 / 16.5 |
> > >
> > > We observed that MSRL generally achieves better performance on the Pets, GTSRB, and DTD datasets when using the combination of all three backbones compared to the other combinations. However, in some cases, MSRL with the three-backbone combination performs worse than the two-backbone setting in (1). This empirical evidence further suggests that increasing the number of views is not always beneficial for specific downstream tasks.
> > >
> > > Response to Q2:
> > >
> > > Re: We agree that a broader comparison strengthens the paper. MSRL represents an emerging paradigm: leveraging foundation models for unsupervised transfer, which is different from traditional MVC methods. Fully unsupervised transfer in multiview learning has attracted increasing attention over the past two years. We included three additional recent baselines: BONE [1] (AAAI 2026), CLUDI [2] (ICML 2025), and CPP [3] (ICLR 2024).
> > >
> > > (1) BONE attempts to address modal differences via view-specific representations.
> > >
> > > (2) CLUDI and CPP utilize pretrained models for self-supervised clustering.
> > >
> > > In addition, we have removed SCMVC and MCMC from the set of competing methods. The experimental results (ACC/NMI/ARI) on all datasets are reported in Table 1 (available at the anonymous link: https://anonymous.4open.science/api/repo/PIC/file/clu.png?v=3c1f288d )
> > >
> > > MSRL consistently outperforms these new baselines across all evaluated datasets.
> > >
> > > [1] H. Xu, et al., Bridging optimization and neural networks for efficient multi-view clustering, AAAI 2026.
> > >
> > > [2] R. Uziel, et al., Clustering via self-supervised diffusion, ICML 2025.
> > >
> > > [3] T. Chu, et al. Image clustering via the principle of rate reduction in the age of pretrained models, ICLR 2024.
> > >
> > > Many thanks again.

---

### Official Review · Reviewer_n5f3 · 2026-03-12

**Soundness:** 3
**Presentation:** 2
**Significance:** 3
**Originality:** 2
**Overall Recommendation:** 4
**Confidence:** 4

**Summary:**

This paper proposes MSRL, an unsupervised transfer learning method that learns invariant representations from heterogeneous features extracted by multiple pretrained visual models. The main idea is to aggregate complementary information within each view and encourage consistent cluster assignments across different views. Specially, the method introduces an information-passing mechanism and an assignment probability distribution consistency scheme. Besides, the model is trained with pseudo-label supervision, cluster diversity regularization, and cross-view consistency constraints. Experiments on eight visual datasets show strong clustering performance under both self-supervised learning and zero-shot transfer learning settings.

**Compliance With Llm Reviewing Policy:**

Affirmed.

**Final Justification:**

The rebuttal has fully resolved my concerns, thus suggesting to accept this paper.

**Key Questions For Authors:**

1、As mentioned in the weaknesses, have the authors considered including clustering visualizations (e.g., t-SNE) to illustrate the effectiveness of the learned invariant representations?

2、The experiments mainly involve ViT-based pretrained models. Have the authors analyzed how the method performs with other backbone types, especially across models with different architectures or representation granularity?

**Limitations:**

No. The paper does not explicitly discuss limitations, but it provides a brief analysis within ViT-based backbones, additional experiments would help better support this point.

**Strengths And Weaknesses:**

Strengths
1、Meaningful Motivation: Instead of assuming homogeneous feature spaces or relying on direct feature fusion, the paper addresses a more realistic setting where features extracted by different pretrained models are heterogeneous. The idea of exploiting complementary information across views through adaptive multiview learning is reasonable and potentially beneficial for robustness and generalization.

2、Theoretical support: The paper provides theoretical analysis and detailed proofs, which help justify the proposed method and make it more grounded.

3、Simple and effective framework: The overall method is relatively simple and easy to understand. The combination of feature aggregation and cross-view consistency is intuitive and effective.

Weaknesses
1、Ambiguous formulation: The method is presented as self-representation learning, but in practice it looks more like an attention-based aggregation design. This makes me wonder whether it introduces some ambiguity.

2、Writing problems: There are some notation inconsistencies in the paper. For example, in Sec. 3.1 the paper refers to M pretrained models but later uses l-indexed models, and the symbols i,j are first used to denote different views but are later reused in Sec. 3.3 to represent different samples within the same view. This makes the method description somewhat confusing. Besides, the logical flow and causal explanations in the writing should be improved to make the paper easier to understand.

3、Insufficient analysis: The paper emphasizes the importance of complementary information, but it would be more convincing to include visualizations that directly show the effectiveness of the learned invariant representations. In addition, the experiments lack experiments on a wider range of pretrained backbones.

---

> ### Author Rebuttal · Authors · 2026-03-29
>
> 1 Ambiguous formulation …self-representation learning, but …like an attention-based aggregation…whether it introduces some ambiguity.
>
> Re: (1) Self-representation learning (SRL) and attention-based aggregation (ABA) in the paper are the theoretical objective and its specific implementation, respectively. As defined in Eq. (3), $a_i$ denotes a coefficient vector. The key difference from traditional representation learning approaches (e.g., sparse or low-rank representation) lies in how the coefficient vector $a_i$ is learned. Instead of solving an algebraic optimization problem, we learn $a_i$ in a data-driven manner. ABA in Eq. (4) is viewed as an implementation tool to achieve the objective of self-representation learning. We will clarify it in Sec. 3.3.
>
> (2) Although ABA shares a similar form with graph attention networks (GAT), their assumptions differ fundamentally. GAT relies on a predefined graph, while MSRL learns sample relationships from all samples guided by encouraging consistent cluster assignments. Even if the coefficient matrix degenerates to the identity matrix, MSRL can still work due to SRL, whereas GAT fails without a graph structure. We will clarify this distinction in the revision.
>
> 2 Writing problems: …notation inconsistencies…M pretrained models…symbols i, j are first used...Besides, the logical flow and causal explanations…should be improved...
>
> Re: (1) $ M $ will be corrected to $L$.
>
> (2) The symbols $i$ and $j$ in Sec. 3.1 will be replaced with alternative symbols $u$ and $v$.
>
> (3) We will enhance the SRL-to-ABA logic as discussed in Q1. Additionally, we will strengthen the causal explanations among SRL, the information-passing mechanism (MPM) and ABA. SRL defines the core problem formulation of MSRL. MPM serves as an interpretive perspective of this formulation. SRL in MSRL can be interpreted as information propagation among samples. ABA is a concrete implementation of SRL for learning $a_i$.
>
> 3 ...including clustering visualizations (e.g., t-SNE) to...invariant representations?
>
> Re: We employed t-SNE visualizations on the PETS, DTD, and AIRCRAFT datasets. We considered three levels of features: the original features from the two pretrained models, the linear features after the linear layers, and the aggregation features generated by the self-attention layers. These figures are provided in the anonymous link: https://anonymous.4open.science/api/repo/PIC/file/vis.jpg?v=fff4d47d
>
> (1) Enhanced Separability: The aggregation features exhibit more compact intra-class clusters and clearer inter-class boundaries.
>
> (2) Invariance Verification: The aggregation features from different views of the same category cluster more distinctly, demonstrating that MSRL captures view-invariant representations.
>
> These results demonstrate that MSRL improves clustering structure and learns view-invariant representations. The visualizations will be included in the experimental section.
>
> 4 …involve ViT-based pretrained models…how...performs with other backbone…
>
> Re: We included ConvNeXt V2 to evaluate generalization beyond ViT-based models. These backbones exhibit different representation granularities: transformer-based models (e.g., DINOv2 and CLIP) primarily produce global semantic representations, while convolutional models (e.g., ConvNeXt V2) capture more localized spatial features. We considered three combinations:
> (1) DINOv2 + ConvNeXt V2
> (2) ViT-L/14 + ConvNeXt V2
> (3) DINOv2 + ViT-L/14 + ConvNeXt V2.
>
> The experimental results (ACC/NMI/ARI) on all datasets were reported in the table.
>
> | Method | Pets | GTSRB | DTD | Aircraft | Flowers | CIFAR-10 | CIFAR-100 | ImageNet-1K |
> | :--- | :--- | :--- | :--- | :--- | :--- | :--- | :--- | :--- |
> | (1) | 92.4 / 93.6 / 87.5 | 30.7 / 46.4 / 20.6 | 55.4 / 65.9 / 41.0 | 25.4 / 53.3 / 15.8 | 99.5 / 99.6 / 99.2 | 98.8 / 97.2 / 97.6 | 79.4 / 85.7 / 69.7 | 67.8 / 85.8 / 56.2 |
> | (2) | 78.5 / 85.8 / 69.6 | 47.6 / 65.2 / 40.7 | 50.2 / 61.9 / 36.3 | 30.8 / 60.2 / 20.1 | 75.1 / 87.2 / 68.9 | 97.3 / 93.3 / 94.1 | 60.0 / 71.7 / 47.2 | 52.9 / 79.8 / 30.9 |
> | (3) | **92.9** / **94.4** / **87.6** | **51.4** / **70.6** / **43.3** | **60.7** / **70.1** / **45.1** | **38.2** / **60.9** / 4.3 | 95.9 / 97.5 / 90.4 | 99.0 / 97.2 / 97.7 | 81.4 / 85.3 / 70.4 | 69.2 / 85.0 / 16.5 |
> | TURTLE | 92.6 / 93.8 / **87.6** | 45.0 / 65.0 / 36.9 | 57.6 / 68.4 / 42.7 | 36.8 / 60.8 / **25.8** | **99.6** / **99.7** / **99.5** | **99.5** / **98.6** / **98.9** | **89.1** / **91.4** / **82.5** | **70.1** / **86.6** / **58.4** |
>
> We observed that MSRL often achieves the encouraging clustering results across different backbone types compared with TURTLE. For example, MSRL outperforms TURTLE on the Pets, GTSRB and DTD datasets when employing the combination of the three backbones. This confirms that the assignment probability distribution consistency scheme effectively captures complementary information even between distinct distributions. The results will be added to the Ablation Study.
>
> Many thanks.

---

> > ### Author Rebuttal · Reviewer_n5f3 · 2026-04-05
> >
> > The rebuttal has fully resolved my concerns, accordingly I will raise my score.

---

> > > ### Author Response · Authors · 2026-04-05
> > >
> > > We sincerely thank the reviewer for their time and effort.

---

### Decision · Program_Chairs · 2026-04-30

**Decision:**

Accept (regular)

**Comment:**

This paper proposes MSRL, a new multiview self-representation learning framework for unsupervised transfer learning. From the review comments, the paper contains the following strengths:

(1) The method combines intuitive components (attention-based aggregation, cross-view consistency, cluster diversity regularization) into a clean and computationally efficient framework that outperforms strong baselines with significant training time reductions.

(2) The paper provides a rigorous theoretical analysis, including a view-incremental monotonic entropy reduction theorem that formally characterizes how clustering quality evolves with additional views, theoretically explaining why increasing the number of views is not always beneficial.

After the rebuttal phase, the paper receives three acceptance recommendations with high confidence scores and one weak rejection recommendation with a low confidence score. Despite some concerns regarding presentation clarity and baseline comparisons, most reviewers with high confidence scores recognize the technical novelty, solid theoretical grounding, and practical relevance of the work. Based on the current recommendation scores and considering the strengths, the paper is suggested to be accepted.